



# High-resolution U.S. methane emissions inferred from an inversion of 2019 TROPOMI satellite data: contributions from individual states, urban areas, and landfills

Hannah Nesser[1], Daniel J. Jacob[1], Joannes D. Maasakkers[2], Alba Lorente[2], Zichong Chen[1], Xiao Lu[3], Lu Shen[4], Zhen Qu[5], Melissa P. Sulprizio[1], Margaux Winter[1], Shuang Ma[6], A. Anthony Bloom[6], John R. Worden[6], Robert N. Stavins[7], Cynthia A. Randles[8*]

[1]School of Engineering and Applied Sciences, Harvard University, Cambridge, MA, USA
[2]SRON Netherlands Institute for Space Research, Leiden, the Netherlands
[3]School of Atmospheric Sciences, Sun Yat-sen University, Zhuhai, Guangdong Province, China
[4]Department of Atmospheric and Oceanic Sciences, School of Physics, Peking University, Beijing, China
[5]Department of Marine, Earth and Atmospheric Sciences, North Carolina State University, Raleigh, NC, USA
[6]Jet Propulsion Laboratory, California Institute of Technology, Pasadena, California, USA
[7]Harvard Kennedy School, Cambridge, MA, USA
[8]ExxonMobil Technology and Engineering Company, Annandale, NJ, USA
[*]Now at United Nations Environment Program International Methane Emissions Observatory, Paris, France

*Correspondence to*: Hannah Nesser (hnesser@g.harvard.com)

**Abstract.** We quantify 2019 methane emissions in the contiguous U.S. (CONUS) at 0.25° × 0.3125° resolution by inverse analysis of atmospheric methane columns measured by the Tropospheric Monitoring Instrument (TROPOMI). A gridded version of the U.S. Environmental Protection Agency (EPA) Greenhouse Gas Emissions Inventory (GHGI) serves as the basis for the prior estimate for the inversion. We optimize emissions and quantify observing system information content for an eight-member inversion ensemble through analytical minimization of a Bayesian cost function. We achieve high resolution with a reduced-rank characterization of the observing system that optimally preserves information content. Our optimal (posterior) estimate of anthropogenic emissions in CONUS is 30.9 (30.0 - 31.8) Tg a$^{-1}$, where the values in parentheses give the spread of the ensemble. This is a 13% increase from the 2023 GHGI estimate for CONUS in 2019. We find livestock emissions of 10.4 (10.0 - 10.7) Tg a$^{-1}$, oil and gas of 10.4 (10.1 - 10.7) Tg a$^{-1}$, coal of 1.5 (1.2 - 1.9) Tg a$^{-1}$, landfills of 6.9 (6.4 - 7.5) Tg a$^{-1}$, wastewater of 0.6 (0.5 - 0.7), and other anthropogenic sources of 1.1 (1.0 - 1.2) Tg a$^{-1}$. The largest increase relative to the GHGI occurs for landfills (51%), with smaller increases for oil and gas (12%) and livestock (11%). These three sectors are responsible for 89% of posterior anthropogenic emissions in CONUS. The largest decrease (28%) is for coal. We exploit the high resolution of our inversion to quantify emissions from 73 individual landfills, where we find emissions are on median 77% larger than the values reported to the EPA's Greenhouse Gas Reporting Program (GHGRP), a key data source for the GHGI. We attribute this underestimate to overestimated recovery efficiencies at landfill gas facilities and to under-accounting of site-specific operational changes and leaks. We also quantify emissions for the 48 individual states in CONUS, which we compare to the GHGI's new state-level inventories and to independent state-produced inventories. Our posterior emissions are on average 34% larger than the 2022 GHGI in the largest 10 methane-producing states, with the biggest upward adjustments in states with large oil and gas emissions, including Texas, New





Mexico, Louisiana, and Oklahoma. We also calculate emissions for 95 geographically diverse urban areas in CONUS. Emissions for these urban areas total 6.0 (5.4 - 6.7) Tg a$^{-1}$ and are on average 39 (27 - 52) % larger than a gridded version of the 2023 GHGI, which we attribute to underestimated landfill and gas distribution emissions.

## 1 Introduction

All projected pathways that prevent global warming above 1.5°C require methane emissions reductions (IPCC, 2022). The Global Methane Pledge, launched at a 2021 meeting of the United Nations Framework Convention on Climate Change (UNFCCC), aims to achieve a 30% global reduction in methane emissions from 2020 to 2030 (About the Global Methane Pledge, 2023). The U.S. government has set goals to decrease methane emissions from landfills by 30% relative to 2015 levels by 2025 and regulation in development aims to reduce oil and gas methane emissions by 30% from 2020 to 2030 (The

White House, 2021). The UNFCCC requires member parties to report their anthropogenic methane emissions including sectoral contributions from oil and gas, coal, livestock, rice, landfills, and wastewater. The bottom-up approaches used to generate these emission inventories use information on sectoral activity levels and emission factors, but considerable uncertainty can exist in these values. Top-down evaluations of bottom-up inventories use observations of atmospheric methane to infer emissions, often through inverse analyses using a chemical transport model. These top-down emission

estimates are most useful if they achieve high spatial resolution and maximize the information content of the observation-model system. Here we use column methane observations from the Tropospheric Monitoring Instrument (TROPOMI) aboard the Sentinel-5 Precursor satellite in a reduced-rank analytical inversion to infer methane emissions and the associated information content at 0.25° × 0.3125° (≈25 km × 25 km) resolution over the contiguous U.S. (CONUS) for 2019, allowing for detailed analysis of sectoral, state, and urban emissions.


Satellite observations of atmospheric methane column concentrations inferred from measurement of backscattered sunlight in the shortwave infrared have been used extensively in inverse analyses of methane emissions (Streets et al., 2013; Jacob et al., 2022). Previous satellite instruments were limited by large pixel sizes (SCIAMACHY, 2003 - 2012) or sparse observations (GOSAT, 2009 - present). TROPOMI provides daily, global observations of atmospheric methane columns at

5.5 km × 7 km nadir pixel resolution (Hu et al., 2018) with a ~3% success rate limited by cloud cover, optically dark surfaces, and heterogeneous terrain (Hasekamp et al., 2019). Inversions of TROPOMI data allow for high-resolution quantification of methane emissions but require understanding the information content of the observations.

Inverse analyses optimize methane emissions (the state vector) by fitting observations to simulated concentrations from a

chemical transport model (CTM) that serves as the inversion forward model. The optimization is typically done by minimizing a Bayesian cost function regularized by a prior emission estimate given by a bottom-up inventory. When a linear relationship exists between emissions and concentrations, as in the case of methane, the optimal (posterior) solution and the



associated error covariances and information content can be found analytically (Brasseur and Jacob, 2017). However, this requires the computationally expensive but embarrassingly parallel construction of the Jacobian matrix that represents the

relationship between emissions and concentrations in the CTM. This matrix is typically constructed by conducting a CTM perturbation simulation for each optimized emission element, limiting either the spatial resolution of the optimized emissions or the size of the inversion domain. Nesser et al. (2021) demonstrated an alternative method that approximates the Jacobian matrix by perturbing emission patterns that are optimally informed by both the prior emissions and the observations. This approach optimally exploits the information content of the observations, quantifying emissions at the highest resolution

possible where the satellite-model observing system provides a constraint and defaulting to the prior estimate elsewhere.

Many inverse studies that quantified U.S. methane emissions using surface, aircraft, or satellite observations have found large discrepancies with the U.S. Environmental Protection Agency's (EPA) Greenhouse Gas Emissions Inventory (GHGI), which is the bottom-up emission estimate reported by the U.S. to the UNFCCC (EPA, 2022a, 2023). Wecht et al. (2014a)

found livestock emissions 40% larger than the GHGI for the summer of 2004. Miller et al. (2013) inferred emissions 50% larger than the GHGI for 2007 and 2008, which they attributed to underestimated oil, gas, and livestock emissions. Turner et al. (2015) found similar results for 2009 to 2011. Maasakkers et al. (2021) inferred oil and gas emissions 35% and 22% higher than the GHGI, respectively, for 2010 to 2015. Lu et al. (2022) found mean 2010 - 2017 anthropogenic emissions 42% larger than the GHGI, which they attributed largely to oil and gas emissions.


Higher resolution regional studies have targeted specific aspects of U.S. methane emissions, including contributions from different sectors, states, and urban areas. Karion et al. (2015) found oil and gas emissions in the Barnett Shale in eastern Texas that were consistent with the GHGI when scaled by the region's relative contribution to national gas production but larger than reported by most basin facilities to the EPA's Greenhouse Gas Reporting Program (GHGRP). A series of studies

inferred much higher emissions in the Permian Basin than implied by a spatially allocated (gridded) version of the GHGI (Zhang et al., 2020; Schneising et al., 2020; Liu et al., 2021; Y. Chen et al., 2022; Varon et al., 2022). Z. Chen et al. (2018) and Yu et al. (2021) found underestimated livestock emissions in the gridded GHGI in the upper Midwest. Jeong et al. (2016) inferred California emissions 20% to 80% larger than a state inventory from the California Air Resources Board (CARB). Plant et al. (2019) found methane emissions from six East Coast urban areas in 2012 to be more than two times

larger than the gridded GHGI.

Here we use the reduced-rank method of Nesser et al. (2021) in an analytical inversion of 2019 TROPOMI observations to quantify emissions at 0.25° × 0.3125° resolution over North America using national emission inventories reported by the U.S., Mexico, and Canada to the UNFCCC as prior estimates. The reduced-rank approach decreases computational cost by

an order of magnitude compared to conventional methods while maximizing information content from TROPOMI. We focus our analysis on CONUS, with particular attention to emissions from individual landfills, states, and urban areas. We compare





our results to the 2023 GHGI (EPA, 2023) and to new emission estimates for individual states published most recently with the 2022 GHGI (EPA, 2022b). Our inversion provides the first observational evaluation for these state inventories. We also compare our results to inventories prepared by individual states and cities.

## 2 Data and methods

We conduct an ensemble of inversions of 2019 TROPOMI methane observations over the North American domain shown in Fig. 1 (9.75°N - 60°N, 130°W - 60°W) using the nested GEOS-Chem CTM at 0.25° × 0.3125° resolution as forward model. The $m = 2919358$ TROPOMI observations are fit to simulated GEOS-Chem concentrations to optimize mean methane emissions for 2019 at the 0.25° × 0.3125° GEOS-Chem resolution. This corresponds to $n = 23691$ emission grid cells with prior methane emissions larger than 0.1 Mg km$^{-2}$ a$^{-1}$, accounting for over 99% of North American methane emissions. In a subset of the ensemble, we optimize boundary conditions for the nested GEOS-Chem simulation for each of the four cardinal directions (north, south, east, and west). Methane chemical and soil sinks are not optimized because they are relatively uniform and slow compared to the ventilation timescale of the domain.

### 2.1 Reduced-rank analytical inversion

The inversion uses $m$ observed concentrations arranged in a vector $\mathbf{y}$ to optimize $n$ gridded emissions arranged in the state vector $\mathbf{x}$ by minimizing a Bayesian cost function $J$ assuming normal errors and regularized by the prior emission estimate $\mathbf{x}_A$ (Rodgers, 2000):

$$J(\mathbf{x}) = (\mathbf{x} - \mathbf{x}_A)^T \mathbf{S}_A^{-1}(\mathbf{x} - \mathbf{x}_A) + \gamma(\mathbf{y} - \mathbf{Kx})^T \mathbf{S}_O^{-1}(\mathbf{y} - \mathbf{Kx}). \tag{1}$$

The prior and observing system error covariance matrices $\mathbf{S}_A$ and $\mathbf{S}_O$, respectively, are assumed diagonal in the absence of better information. The regularization factor $\gamma$ corrects for the absence of covariance in $\mathbf{S}_O$ (Chevallier, 2007). We generate an eight-member inversion ensemble using a range of prior error variances and $\gamma$ values to capture the inversion's sensitivity to uncertainty in these parameters (Sect. 2.7). The reduced-rank Jacobian matrix $\mathbf{K} = \partial\mathbf{y}/\partial\mathbf{x}$ represents the sensitivity of concentrations to emissions in the CTM. We construct a rank-$k$ Jacobian matrix for the 0.25° × 0.3125° GEOS-Chem grid by perturbing in the CTM the $k$ emission patterns that best capture the prior emissions and the information content of the TROPOMI observations (Sect. 2.6).

Analytical minimization of the cost function following Rodgers (2000) yields the optimal (posterior) state vector estimate $\hat{\mathbf{x}}$, error covariance matrix $\hat{\mathbf{S}}$, and information content given by the averaging kernel matrix $\mathbf{A} = \partial\hat{\mathbf{x}}/\partial\mathbf{x} = \mathbf{I} - \hat{\mathbf{S}}\mathbf{S}_A^{-1}$, which describes the sensitivity of the posterior estimate to the true state vector. However, this solution requires inverting the cost function Hessian, which produces numerical instabilities due to the rank reduction of the Jacobian matrix. Here we use a



reduced-rank approximation of the posterior solution following Bousserez and Henze (2018) to solve the inversion on an orthonormal basis that optimally spans the information content of the satellite–forward model observing system. The basis is given by the eigendecomposition of the prior-preconditioned Hessian of the cost function,

$$\widehat{\mathbf{H}}_p = \mathbf{S}_A^{1/2} \mathbf{K}^T \mathbf{S}_O^{-1} \mathbf{K} \mathbf{S}_A^{1/2} = \mathbf{V} \mathbf{\Lambda} \mathbf{V}^T, \tag{2}$$

where the columns of $\mathbf{V}$ are the eigenvectors and $\mathbf{\Lambda}$ is a diagonal matrix with entries equal to the eigenvalues. The calculation of $\widehat{\mathbf{H}}_p$ requires substantial memory for large $m$ and $n$, for which we use Dask, a Python parallelization package (Dask Development Team, 2016). The reduced-rank posterior approximation is then generated using the largest $k$ eigenvalues $\mathbf{\Lambda}_k$ and the associated eigenvectors $\mathbf{V}_k$ (Bousserez and Henze, 2018):

$$\mathbf{A_K} = \gamma \mathbf{S}_A \mathbf{V}_k \mathbf{\Lambda}_k (\mathbf{I}_k + \gamma \mathbf{\Lambda}_k)^{-1} \mathbf{V}_k^T \mathbf{S}_A, \tag{3}$$

$$\widehat{\mathbf{S}}_\mathbf{K} = (\mathbf{I}_n - \mathbf{A_K}) \mathbf{S}_A, \text{and} \tag{4}$$

$$\widehat{\mathbf{x}}_{FR} = \mathbf{x}_A + \gamma \widehat{\mathbf{S}}_\mathbf{K} \mathbf{K}^T \mathbf{S}_O^{-1} \big( \mathbf{y} - \mathbf{F}(\mathbf{x}_A) \big). \tag{5}$$

Here, $\widehat{\mathbf{x}}_{FR}$ approximates the full-rank (FR) posterior $\widehat{\mathbf{x}}$ by minimizing the difference between the two, and $\widehat{\mathbf{S}}_\mathbf{K}$ and $\mathbf{A_K}$ are the optimal posterior error covariance and averaging kernel matrices, respectively, for an inversion solved with a reduced-rank forward model. We set $k$ to match the rank of the reduced-rank Jacobian matrix, which is chosen to maximize information content within the available computational resources (Sect. 2.6). The diagonal elements of $\mathbf{A}_k$ are often referred to as averaging kernel sensitivities and are a measure of the dependence of the optimized emissions on the prior estimate. Their sum (trace of $\mathbf{A}_k$) gives the degrees of freedom for signal (DOFS) that represent the number of pieces of information independently quantified by the observing system (Rodgers, 2000). The reduced-rank inversion and Jacobian matrix do not attempt to optimize emissions in areas with low information content, so we default to the prior estimate for grid cells with averaging kernel sensitivities less than 0.05 (Nesser et al., 2021).

## 2.2 Prior estimates and errors

Figure 1 shows the prior emission estimates for different sectors. Anthropogenic emissions are given by the spatially disaggregated (gridded) versions of the 2016 EPA GHGI for the U.S. for 2012 (Maasakkers et al., 2016), the Instituto Nacional de Ecología y Cambio Climático (INECC) inventory for Mexico for 2015 (Scarpelli et al., 2020), and the Environment and Climate Change Canada (ECCC) inventory for Canada for 2018 (Scarpelli et al., 2021). We update the distribution and magnitude of GHGI oil and gas emissions to the 2020 GHGI for 2018 following Shen et al. (2022) and use the Environmental Defense Fund's inventory for the Permian basin for 2019 (Zhang et al., 2020), where GHGI estimates are known to be too low (Zhang et al., 2020; Schneising et al., 2020; Liu et al., 2021; Y. Chen et al., 2022; Varon et al., 2022).



We treat oil and gas as a single sector in our analysis due to significant source co-location and uncertainty in the partitioning of oil and gas wells. The magnitude of GHGI livestock, landfill, and wastewater emissions changed by less than 10% from 2012 to 2019, while coal emissions decreased by 26%. The distribution of these sources is unlikely to have changed significantly. Anthropogenic emissions for Central America and the Caribbean islands are from the EDGAR v4.3.2 global emission inventory for 2012 (Janssens-Maenhout et al., 2019). Anthropogenic emissions are assumed aseasonal except for manure management and rice cultivation, for which we apply monthly scaling factors as described by Maasakkers et al. (2016) and Zhang et al. (2018), respectively.

Prior emissions for wetlands are given by the high-performance subset of the WetCHARTs ensemble version 1.3.1, which includes the nine ensemble members that best match global GOSAT inversion results (Ma et al., 2021). Lu et al. (2022) found in an inversion of GOSAT data over North America that this high performance subset overestimated wetland methane emissions, particularly at high latitudes. We remove from the ensemble the two members (WetCHARTs models 1923 and 2913; Bloom et al., 2017) that are most responsible for this overestimate. Other natural methane emission sources are minor and include open fires, termites, and geological seeps, for which we follow the emissions described in Lu et al. (2022). Methane losses from chemical reaction, soil uptake, and stratospheric oxidation are prescribed as in Maasakkers et al. (2019) and are not optimized in the inversion.

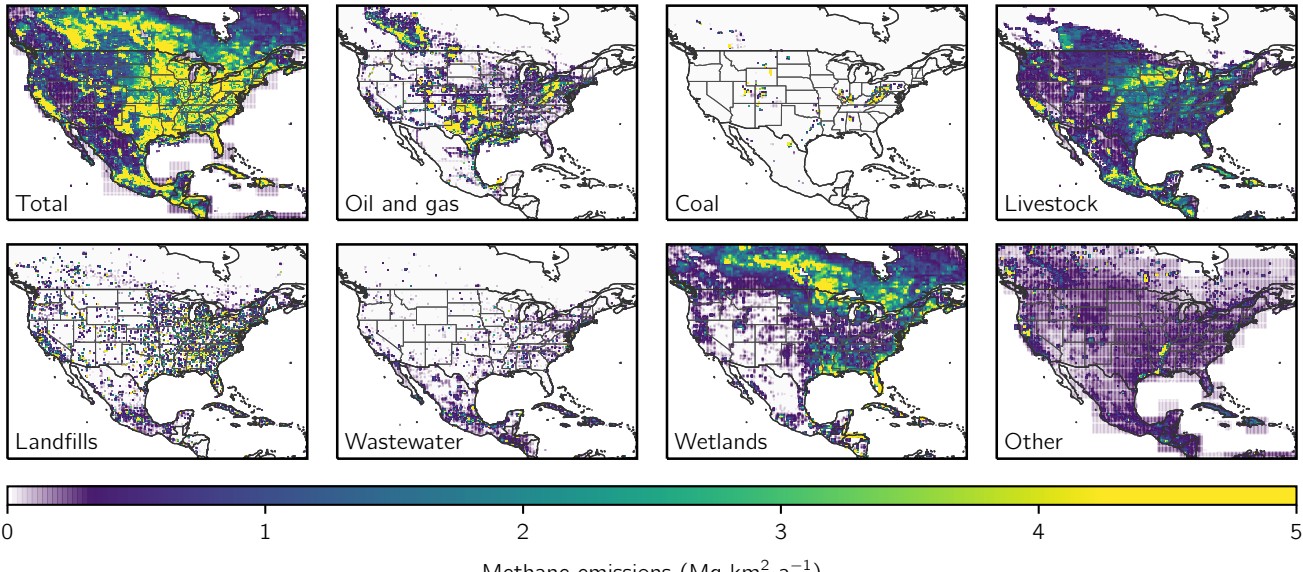

**Figure 1:** Bottom-up methane emission inventories used as prior estimates for the inversion. Panels show annual mean methane emissions for different sectors. Anthropogenic sectors are given by the gridded versions of the national inventories of Canada (ECCC), the U.S. (EPA GHGI), and Mexico (INECC) reported to the UNFCCC (Maasakkers et al., 2016; Scarpelli et al., 2020, 2022). U.S. oil and gas emissions are updated as described in Sect. 2.2. Wetland emissions are given by the high-performance subset of the WetCHARTs version 1.3.1 wetlands inventory ensemble (Ma et al., 2021), excluding two ensemble members as described in Sect. 2.2. Emissions are shown on the 0.25° × 0.3125° GEOS-Chem grid used for the inversion.



We assume uniform relative error standard deviations for the prior emissions of between 50% and 100% for the different members of our inversion ensemble, with no error covariance between grid cells. Previous inversions that optimized methane emissions over North America assumed prior error standard deviations up to 50% . We inflate errors up to 100% in our ensemble to account for increased errors at high resolution (Maasakkers et al., 2016). Errors for each ensemble member are chosen as described in Sect. 2.7.

## 2.3 Forward model

We use the nested version of the GEOS-Chem CTM 12.7.1 (DOI: 10.5281/zenodo.3676008) at 0.25° × 0.3125° resolution over North America as the forward model for the inversion. Earlier versions of the methane simulation were described by Wecht et al. (2014a) and Turner et al. (2015). The model is driven by GEOS-FP meteorological fields from the NASA Global Modeling and Assimilation Office (Lucchesi, 2017). Methane sinks from OH, Cl, soil uptake, and stratospheric oxidation are as described in Maasakkers et al. (2019). Initial conditions for January 1, 2019 and 3-hourly boundary conditions for the year are specified by methane concentration fields from a global GEOS-Chem simulation at 2° × 2.5° resolution using optimized emissions from a global inversion of TROPOMI observations (Qu et al., 2021).

## 2.4 TROPOMI observations

TROPOMI has provided daily, global observations of dry column methane mixing ratios at 7 km × 7 km nadir pixel resolution since May 2018 and at 5.5 km × 7 km nadir pixel resolution since August 2019 (Lorente et al., 2021). TROPOMI measures backscattered solar radiation in the 2.3 μm methane absorption band from a sun-synchronous orbit with a local overpass time of 13:30 (Veefkind et al., 2012). Methane concentrations are inferred from a full-physics retrieval with a ~3% success rate limited by cloud cover, variable topography, low or heterogeneous albedo, and high aerosol loading (Hasekamp et al., 2019). We use retrieval v14 as described by Lorente et al. (2021), which has a -3.4 ± 5.6 ppb bias relative to the Total Carbon Column Observing Network (TCCON). We use only high-quality retrievals as indicated by the quality assessment flag.

Previous analyses of TROPOMI data identified surface artifacts (Barré et al., 2021) and spatially variable biases relative to the more accurate but sparser GOSAT data (Qu et al., 2021; Z. Chen et al., 2022). We filter the data to remove snow- and ice-covered scenes using blended albedo, an empirical parameter developed by Wunch et al. (2011) and suggested for the TROPOMI data by Lorente et al. (2021). We remove scenes with blended albedo greater than 0.75 in non-summer seasons. We also remove scenes with albedo in the shortwave infrared less than 0.05 following de Gouw et al. (2020), which account for most of the remaining unphysical TROPOMI observations (methane mixing ratio less than 1700 ppb), and scenes north of 50°N in winter.

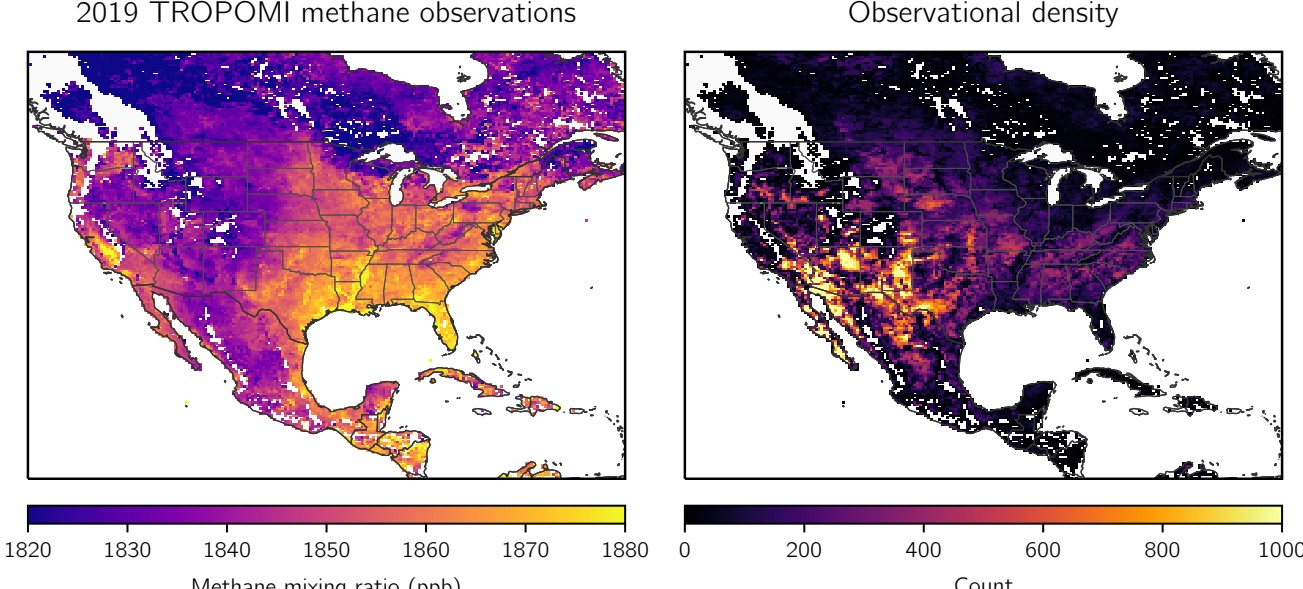

**Figure 2:** TROPOMI methane observations in 2019. The left panel shows the annual average column dry methane mixing ratios for 2019 averaged on the 0.25° × 0.3125° GEOS-Chem grid. The right panel shows the number of observations for the year on the same grid. The observations have been filtered as described in Sect. 2.4.

Figure 2 shows the final $m$ = 2919358 observations used for the inversion on the GEOS-Chem 0.25° × 0.3125° grid. The filters preserve 69% of the high-quality retrievals of TROPOMI v14 and increase the GOSAT - TROPOMI correlation in all seasons, with the largest increases in winter and spring (Fig. S1). Seasonal regional biases decrease by between 7% and 21% and are always within the one standard deviation range of both the TROPOMI and GOSAT data. Comparison to a GEOS-

225 Chem simulation driven by the prior emissions as shown in Fig. S2 shows a mean aseasonal (GEOS-Chem - TROPOMI) bias of ξ = 9.1 ppb over North America which we attribute to errors in the boundary conditions. This bias can also be fit as a linear function of degrees latitude θ as ξ = −5.40 + 0.39θ. We correct the bias in our inversion ensemble members by removing either the continental mean bias or the latitude-dependent correction from the GEOS-Chem concentrations.

### 2.5 Observing system errors

The observing system error covariance matrix $\mathbf{S}_O$ includes contributions from forward model, instrument, and representation errors (Brasseur and Jacob, 2017).We calculate the total observing system error variances using the residual error method (Heald et al., 2004). This method assumes that the mean difference between the TROPOMI observations and the prior GEOS-Chem simulation, calculated here on a seasonal 2° × 2° grid, is caused by errors in emissions that will be corrected by the inversion. The standard deviation of the residual errors after subtracting the mean gridded errors then defines the

standard deviation of the observing system errors. We set a minimum error standard deviation of 10 ppb, which applies to 32% of observations. We find a mean observing system error standard deviation of 11.5 ppb, with the largest errors in winter



and at high latitudes. The resulting error variances are the diagonal elements of $\mathbf{S}_O$. Off-diagonal terms are assumed zero in the absence of better information, which we account for with the regularization factor $\gamma$ (Chevallier, 2007). We describe the choice of $\gamma$ in Sect. 2.7.

**2.6 Jacobian matrix**

Constructing the Jacobian matrix $\mathbf{K}$ for our inversion would normally require conducting a 1-year perturbation simulation for each of the $n$ = 23691 grid cells optimized. This is computationally intractable. We construct the Jacobian matrix at substantially decreased computational cost using the reduced-rank method introduced by Nesser et al. (2021), which takes advantage of the heterogeneous information content of the TROPOMI observations. This method updates an initial, low-cost 245 estimate of the Jacobian matrix by perturbing the patterns that best explain the information content of the observing system, constructing a reduced-rank Jacobian matrix that optimally preserves information content.

We construct the initial, low-cost estimate of the Jacobian matrix $\mathbf{K}^{(0)}$ using the mass-balance approach described by Nesser et al. (2021). We assume that a perturbation of methane emissions $\Delta x_j$ in grid cell $j$ produces column mixing ratio 250 enhancements $\Delta y_i$ over grid cell $i$ according to

$$\Delta y_i = \alpha_{ij} \frac{M_{\text{air}}}{M_{\text{CH}_4}} \frac{Lg}{Up} \Delta x_j \tag{6}$$

where $\alpha_{ij} \in [0, 1]$ is a dimensionless coefficient providing a crude representation of turbulent diffusion, $M_{\text{air}}$ and $M_{\text{CH}_4}$ are 255 the molecular weights of dry air and methane, respectively, $L$ is a ventilation length scale equal to the square root of the grid cell area, $g$ is gravitational acceleration, $U$ is the wind speed taken here as 5 km h$^{-1}$, and $p$ is the surface pressure taken here as 1000 hPa. The use of $\alpha_{ij}$ produces off-diagonal structure in $\mathbf{K}^{(0)}$, which we found in Nesser et al. (2021) to be necessary for an effective first estimate. We apply a simple isotropic turbulent diffusion scheme in which the influence of emissions spreads linearly to concentric rings of grid cells. This is represented as $\alpha_{ij} = (8 - \|i - j\|)/36c$, where $\|i - j\| =$ 260 $\{0, 1, \dots, 7\}$ gives the distance in latitude or longitude grid cell index between $i$ and $j$, 36 is the sum of $\|i - j + 1\|$ values, and $c$ gives the number of grid cells in the corresponding concentric ring. For $\|i - j\| \geq 8$, $\alpha_{ij} = 0$.

We use $\mathbf{K}^{(0)}$ together with the error covariance matrices $\mathbf{S}_A$ and $\mathbf{S}_O$ to calculate the initial patterns of information content that are perturbed in the forward model. We calculate the prior pre-conditioned Hessian (Eq. (2)) using $\mathbf{K}^{(0)}$ and perform its 265 eigendecomposition. The resulting matrix of eigenvectors $\mathbf{V}^{(0)}$ is related to the patterns of information content via $\mathbf{S}_A^{1/2}\mathbf{V}^{(0)}$, which is equivalent to the eigenvector matrix of the averaging kernel matrix calculated with $\mathbf{K}^{(0)}$ (Bousserez and Henze, 2018). We perturb the $k_1$ = 434 eigenvectors that capture 50% of the DOFS generated with $\mathbf{K}^{(0)}$. We then apply an optimal



operator that restores the original state dimension and minimizes information content loss to yield an updated reduced-rank Jacobian matrix estimate $\mathbf{K}^{(1)}$. We recompute the eigenvectors, perturb the $k_2 = 1952$ eigenvectors that explain 80% of the

270 initial DOFS, and construct the final reduced-rank Jacobian matrix $\mathbf{K}^{(2)}$. This iterative update scheme optimizes the information content of the posterior solution while reducing the computational cost by an order of magnitude (Nesser et al., 2021).

## 2.7 Inversion ensemble

The posterior error covariance matrix that results from Bayesian optimization (Eq. (4)) does not account for errors in

inversion parameters including the prior and observing system error covariance matrices (Houweling et al., 2014). The analytical solution readily allows for the creation of an ensemble of inversions that reflects the sensitivity of the results to the chosen setup including parameters. Table 1 summarizes our quality-controlled ensemble of inversions. We conduct inversions that do or do not optimize the boundary conditions and apply either a latitudinal or mean bias correction to the prior (model - observation) difference as driven by boundary condition biases. For each inversion, we choose the relative

prior error (50%, 75%, or 100%) and regularization factor (between 0.175 and 0.5) so that the prior term of the cost function evaluated at the posterior solution $J_A(\hat{\mathbf{x}}) = (\hat{\mathbf{x}} - \mathbf{x}_A)^{\mathrm{T}} \mathbf{S}_A^{-1} (\hat{\mathbf{x}} - \mathbf{x}_A)$ averages to 1 across all grid cells optimized by the

**Table 1:** The eight members of the inversion ensemble.

| Optimized boundary conditions[1] | Bias correction[2] | Prior error standard deviation[3] | Regularization factor[3] |
|---|---|---|---|
| Yes | Latitudinal | 50% | 0.2 |
|  |  | 75% | 0.45 |
| Yes | Mean | 50% | 0.175 |
|  |  | 75% | 0.3 |
|  |  | 100% | 0.5 |
| No | Latitudinal | 50% | 0.175 |
|  |  | 75% | 0.35 |
| No | Mean | 75% | 0.175 |

[1] We conduct inversions that either do or do not optimize the boundary conditions. In inversions with optimized boundary conditions, we
include in the inversion state vector four boundary condition elements corresponding to the northern, eastern, southern, and western borders of the North American domain.
[2] We also conduct inversions that apply either a latitudinal or mean bias correction to the prior (model – observation) difference. The latitudinal correction fits the bias with a first order polynomial. In inversions with a mean bias correction, we remove the mean prior (model – observation) difference as driven by boundary condition biases.
[3] We balance the prior and observing system errors to avoid overfitting the emissions to the observations. The regularization factor $\gamma$ is applied to the inverse observing system error covariance matrix $\mathbf{S}_0^{-1}$ so that values less than one increase the observing system errors. We choose the value of the regularization factor and the prior error standard deviation for a given inversion so that the prior term of the posterior cost function is approximately one as required by chi-squared statistics (Sect. 2.7).





reduced-rank inversion as expected from the chi-square distribution, which $J_A(\mathbf{x})$ definitionally follows (Lu et al., 2021).
This yields an ensemble of eight quality-controlled inversions with indistinguishable validity. All inversions have few grid cells with negative emissions, most of which are on the same order of magnitude as the soil sink. We report the mean posterior emissions for the ensemble, with uncertainty ranges given by the ensemble range.

**2.7 Source attribution**

The high resolution of the inversion facilitates the attribution of the posterior emission estimates to individual source sectors or regions, including states and urban areas. We aggregate the native resolution emission and error estimates to the corresponding $p$ sectors, states, or urban areas using a summation matrix $\mathbf{W} \in \mathbb{R}^{p \times n}$. The rows of $\mathbf{W}$ are given by the relative contribution of each grid cell to each source category. For sectoral attribution, the rows are given by the relative contribution of each grid cell to a given sector in the prior emission estimate. For state attribution, the rows are given by the fraction of each grid cell within a given state. For urban area attribution, the rows have binary values depending on whether the grid cell overlaps with a given urban area. If the grid cell contains multiple urban areas, the fractional contribution of the grid cell to a given urban area is used instead. The reduced-dimension posterior estimate $\hat{\mathbf{x}}_{\text{FR,red}}$, posterior covariance matrix $\hat{\mathbf{S}}_{\text{K,red}}$, and averaging kernel matrix $\mathbf{A}_{\text{K,red}}$ are then given by

$$\hat{\mathbf{x}}_{\text{FR,red}} = \mathbf{W}\hat{\mathbf{x}}_{\text{FR}}, \tag{7}$$
$$\hat{\mathbf{S}}_{\text{K,red}} = \mathbf{W}\hat{\mathbf{S}}_{\text{K}}\mathbf{W}^{\text{T}}, \text{and} \tag{8}$$
$$\mathbf{A}_{\text{K,red}} = \mathbf{W}\mathbf{A}_{\text{K}}\mathbf{W}^{*}, \tag{9}$$

where $\mathbf{W}^{*} = \mathbf{W}^{\text{T}}(\mathbf{W}\mathbf{W}^{\text{T}})^{-1}$ is the Moore-Penrose pseudo inverse (Calisesi et al., 2005). In the case of disaggregating our emission estimates to individual landfills, we scale the posterior estimate in the corresponding grid cell by the fraction of emissions attributed to landfills in the prior estimate. These approaches to source aggregation and disaggregation assume that the prior fractional sectoral contributions are correct in each grid cell and that emission sources are evenly distributed in grid cells that cross state lines. The high resolution of our emission estimates decreases the influence of these assumptions relative to coarser resolution estimates. Newly developed methods use prior and posterior error covariances to improve upon these assumptions (Cusworth et al., 2021).

**3 Results and discussion**

Figure 3 shows the ensemble mean posterior scale factors relative to the prior emission estimate as described in Sect. 2.2 (left) and the corresponding averaging kernel sensitivities (right). Grid cells unoptimized by the inversion (mean averaging kernel sensitivity less than 0.05) are left blank. We find 772 (421 - 1279) DOFS for the domain, where the values in parentheses are the ensemble minimum and maximum, respectively. This represents a large increase in information content

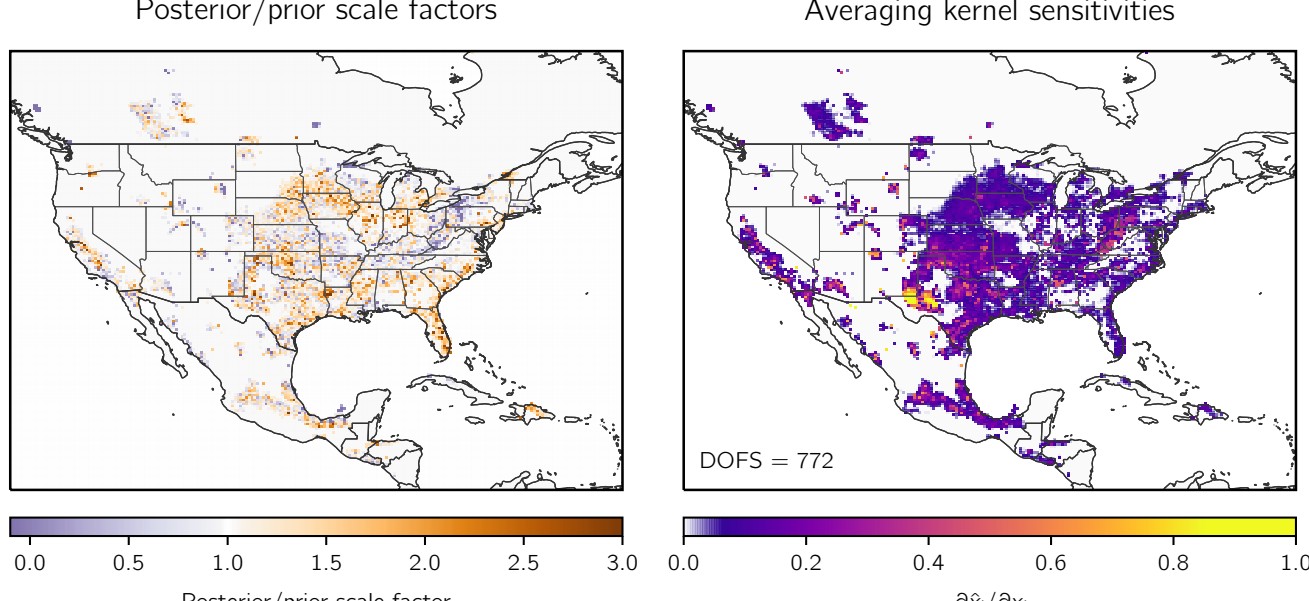

**Figure 3:** Optimization of methane emissions for 2019 by inversion of TROPOMI observations. The left panel shows the scale factors
relative to the prior estimate for the inversion given by gridded versions of the national anthropogenic emissions inventories for the U.S.
(EPA GHGI), Mexico (INECC), and Canada (ECCC), with U.S. oil and gas emissions updated as described in Sect. 2.2, and by
WetCHARTs wetland emissions (top left panel of Fig. 1). The right panel shows the observing system information content as measured by
the averaging kernel sensitivities (the diagonal elements of the averaging kernel matrix). Values of 1 indicate that TROPOMI quantifies
emissions independently of the prior estimate, while values of 0 indicate that emissions are not optimized by the inversion. The sum of the
averaging kernel sensitivities gives the degrees of freedom for signal (DOFS), shown inset, which defines the number of independent
pieces of information quantified by the observing system. Grid cells with averaging kernel sensitivities less than 0.05 are left blank.

relative to past inversions over North America: Lu et al. (2022) found 114 DOFS in a joint inversion of data from GOSAT
and the National Oceanic and Atmospheric Administration's (NOAA) GLOBALVIEWplus ObsPack in situ data, while Shen
et al. (2022) found 201 DOFS in an inversion of TROPOMI observations over 14 oil and gas basins. This increase reflects
both the improved coverage from TROPOMI and the benefit of achieving 0.25° × 0.3125° resolution on the continental
scale. Of these DOFS, 641 (350 – 1058) are found for CONUS, 86 (53 - 134) for Mexico, and 37 (15 - 69) for Canada. The
high information content for CONUS reflects both the large emissions (Fig. 1) and the high density of TROPOMI
observations (Fig. 2). As a result, we focus our discussion on CONUS. We isolate anthropogenic emissions by removing
contributions from wetlands and other natural sources following Sect. 2.8. We compare our posterior emissions to the 2023
EPA GHGI inventory for 2019 (henceforth "GHGI23"; EPA, 2023) and to the most recent emission estimates for individual
states as published with the 2022 EPA GHGI inventory for 2019 (henceforth "GHGI22"; EPA, 2022b). We remove
emissions from Hawaii and Alaska from the GHGI total using the GHGI22 state estimates scaled to match the GHGI23
sectoral totals.



We evaluate the inversion results by comparing simulated observations from GEOS-Chem driven by either the prior or the mean posterior emissions to TROPOMI observations and to independent in situ surface and tower observations from NOAA's GLOBALVIEWplus $CH_4$ ObsPack v3.0 database (Cooperative Global Atmospheric Data Integration Project, 2019). We follow Lu et al. (2021) and use only daytime ObsPack observations with outliers excluded. We use monthly average ObsPack observations over CONUS to increase consistency with the annual temporal resolution of our inversion and

its distribution of information content. Compared to TROPOMI, both the prior and posterior GEOS-Chem simulations produce similar coefficients of determination ($R^2$) and root mean squared errors (RMSEs). Compared to ObsPack, the posterior simulation improves upon the prior simulation, increasing $R^2$ from 0.55 to 0.65 and decreasing the RMSE from 80 ppb to 73 ppb, similar to previous inversions of satellite data (Lu et al., 2021). The broad agreement of both simulations with observations reflects the high quality of the prior emission estimate in North America (Maasakkers et al., 2019). We also

compare the TROPOMI v14 data used here to the most recent data (v19), which has improved bias corrections and performance compared to GOSAT in North America (Balasus et al., 2023). We find no correlation ($R^2 = 0.03$) between our posterior scaling factors and the mean (v14 - v19) difference, suggesting that biases in the v14 data do not influence our posterior emissions.

## 3.1 CONUS sectoral emissions

We find posterior anthropogenic methane emissions of 30.9 (30.0 - 31.8) Tg $a^{-1}$ for CONUS in 2019, a 13% increase from the GHGI23 estimate of 27.3 (25.1 - 30.6) Tg $a^{-1}$, where the values in parentheses represent the GHGI23 95% confidence interval (EPA, 2023). Lu et al. (2022) found larger anthropogenic emissions of 36.2 (32.1 - 37.6) Tg $a^{-1}$ over the same domain for 2017 by optimizing emissions and trends in a joint inversion of GOSAT and in situ observations for 2010 to 2017. Worden et al. (2022) found lower anthropogenic emissions of 27.6 (22.6 – 23.9) Tg $a^{-1}$ over the U.S. for 2019 by

regridding global inversions of GOSAT data that optimized emissions at $2° \times 2.5°$ resolution using uncertainties for the prior and posterior estimates. Deng et al. (2022) reviewed an ensemble of global inversions and found median U.S. posterior anthropogenic emissions for 2019 of 26.5 (20.8 - 38.7) Tg $a^{-1}$ with GOSAT data and 31.9 (23.9 - 43.1) Tg $a^{-1}$ with in situ data.

We allocate our national total to individual emission sectors using the attribution method described in Sect. 2.8. From the off-diagonal structure of $\hat{\mathbf{S}}_{\mathbf{K},\text{red}}$ (Eq. (8)), we find very low posterior error correlation between the sectors (mean error correlation coefficients less than 0.2), indicating that we can accurately separate sectoral emissions. Figure 4 and Table 2 summarize the results compared to the GHGI23. Livestock, oil and gas, and landfills account for 89% of posterior anthropogenic emissions and all increase relative to the GHGI23. We find a significant decrease from the GHGI23 only for

coal. For these four sectors, we find sectoral averaging kernel sensitivities between 0.47 and 0.91, larger than the values found by Lu et al. (2022) from GOSAT and in situ data. We find a small but significant increase in wetland emissions that is consistent with the large range found by Lu et al. (2022). However, the reduced-rank observing system only optimizes about



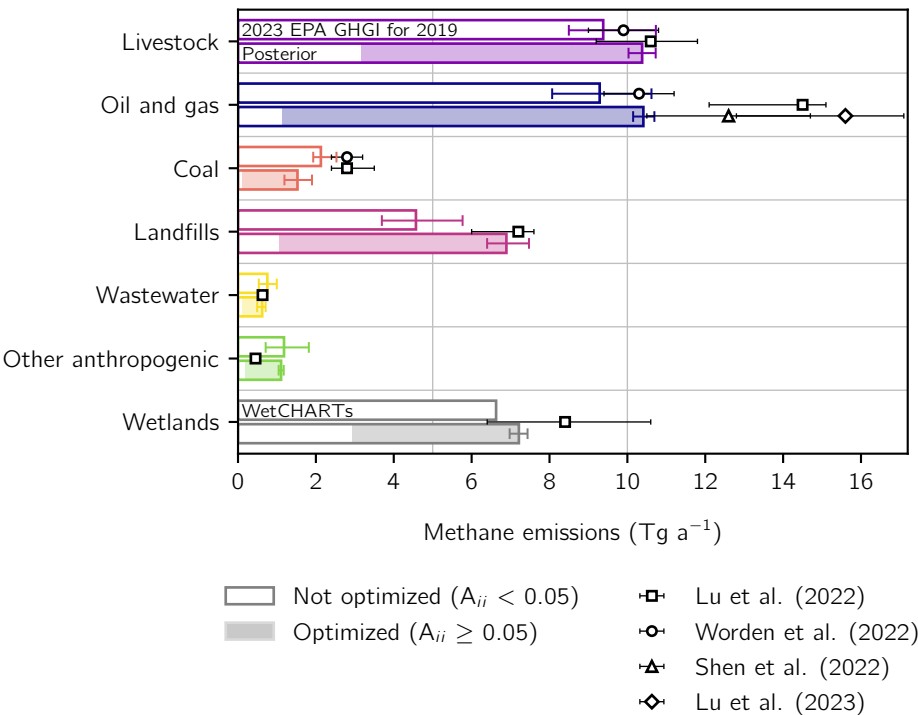

**Figure 4:** Sectoral methane emissions in the contiguous United States (CONUS) for 2019. The 2023 EPA GHGI emissions for 2019 (top
bars) and posterior estimates given by inversion of TROPOMI data for 2019 (bottom bars) are shown for different sectors. For wetland
emissions we show the WetCHARTs estimate (top bar). The shading corresponds to emissions in grid cells that are optimized by the
inversion (grid cells with averaging kernel sensitivities greater than 0.05), while the white represents emissions not optimized by the
inversion so that the posterior defaults to the prior estimate. Error bars on the GHGI emissions represent the GHGI 95% confidence
intervals. Error bars on the posterior emissions are given by the spread of the eight-member inversion ensemble. Also shown are
independent sectoral emission estimates from previous inversions.

half of wetland emissions, with most of the inferred increase limited to the south eastern coast, including South Carolina,

Georgia, and eastern Florida.

Landfill emissions show the largest relative and absolute increase from the GHGI23 for 2019. We find posterior emissions of

6.9 (6.4 - 7.5) Tg a$^{-1}$, a 51% increase relative to the GHGI23 estimate of 4.6 (3.7 - 5.8) Tg a$^{-1}$. Lu et al. (2022) found similar

posterior landfill emissions of 7.5 (5.9 - 7.7) Tg a$^{-1}$ for 2017. We attribute the GHGI23 underestimate to two components of

the GHGRP landfill inventory methodologies that produce key inputs for the GHGI, which we discuss in detail in Sect. 3.2.

First, for landfills with gas recovery systems, the GHGRP assumes too-high collection efficiencies. Second, the GHGRP

does not account for site-specific operations that may produce anomalous emissions.






**Table 2:** 2019 methane emissions for the contiguous United States (CONUS).

|  | Inventory emissions[1] | Posterior emissions[2] | Sensitivity[3] |
|---|---|---|---|
| **Total sources (Tg a$^{-1}$)** | 35.1 | 39.3 (38.2 - 40.3) | |
| **Anthropogenic sources** | 27.3 (25.1 - 30.6) | 30.9 (30.0 - 31.8) | |
| Livestock | 9.4 (8.5 - 10.7) | 10.4 (10.0 - 10.7) | 0.66 (0.55 - 0.76) |
| Oil and natural gas | 9.3 (8.1 - 10.6) | 10.4 (10.1 - 10.7) | 0.91 (0.88 - 0.95) |
| Coal | 2.1 (1.9 - 2.5) | 1.5 (1.2 - 1.9) | 0.60 (0.45 - 0.80) |
| Landfills | 4.6 (3.7 - 5.8) | 6.9 (6.4 - 7.5) | 0.47 (0.34 - 0.64) |
| Wastewater | 0.8 (0.5 - 1.0) | 0.6 (0.5 - 0.7) | 0.33 (0.16 - 0.60) |
| Other anthropogenic | 1.2 (0.7 - 1.8) | 1.1 (1.0 - 1.2) | 0.59 (0.44 - 0.76) |
| **Natural sources** | 7.8 | 8.4 (8.1 - 8.6) | |
| Wetlands | 6.6 | 7.2 (7.0 - 7.4) | 0.35 (0.16 - 0.55) |
| Other biogenic | 1.1 | 1.2 (1.2 - 1.2) | 0.25 (0.19 - 0.32) |

[1]Inventory estimates of sectoral methane emissions. Anthropogenic emissions are given by the EPA 2023 GHGI for 2019, with error ranges inferred from the sum in quadrature of bottom-up subsector errors given as 95% confidence intervals. Wetland emissions are from a subset of the high performance WetCHARTs ensemble version 1.3.1; see Sect. 2.2 for details.

[2]Optimized emissions from the inversion of TROPOMI data, with the range from the eight members of the inversion ensemble shown in parentheses.

[3]The sensitivity of the posterior emissions to the observing system as given by the diagonal elements of the sectoral averaging kernel matrix calculated as described in Sect. 2.8. The values in parentheses give the range of the inversion ensemble. Values range from 0 (no sensitivity) to 1 (full sensitivity).


Coal mining emissions of 1.5 (1.2 - 1.9) Tg a$^{-1}$ exhibit the largest decrease in sectoral emissions relative to the GHGI23 estimate of 2.1 (1.9 - 2.5) Tg a$^{-1}$. Lu et al. (2022) found much larger posterior emissions of 2.9 (2.3 - 3.4) Tg a$^{-1}$ for 2017, and Worden et al. (2022) found similar values of 2.8 ± 0.4 Tg a$^{-1}$ for 2019. Compared to these studies, we achieve a stronger constraint on coal emissions as measured by averaging kernel sensitivities, reflecting the increased coverage from

TROPOMI compared to GOSAT. Our lower estimate better reflects the 30% decrease in CONUS coal production from 2012 to 2019 (EIA, 2021), which is also shown in the 30% decrease in GHGI23 coal emissions over the same period (EPA, 2023). As expected, emissions correlate with underground coal mining: Appalachia generated 56% of U.S. coal from underground mines in 2019 and 64% of posterior emissions from coal, while the Illinois Basin yielded 30% of U.S. underground coal and 20% of posterior emissions (EIA, 2021).


Livestock emissions show broad agreement with the GHGI23, with posterior emissions of 10.4 (10.0 - 10.7) Tg a$^{-1}$ representing an 11% increase from the GHGI23 estimate of 9.4 (8.5 - 10.7) Tg a$^{-1}$. Lu et al. (2022) found similar mean posterior livestock emissions of 10.4 (8.8 - 11.6) Tg a$^{-1}$ over CONUS for 2017, and Worden et al. (2022) found similar values of 9.9 ± 0.4 Tg a$^{-1}$ for 2019. Yu et al. (2021) conducted a seasonal inversion of aircraft observations over the North

Central U.S. and South Central Canada for 2017 to 2018 and found mean posterior livestock emissions of 5.5 (5.1 - 6.2) Tg



a[-1], which agrees with our livestock estimate of 5.4 (5.2 - 5.6) Tg a[-1] over the same region. Despite agreement with total GHGI23 livestock estimates, we find a significant increase in manure management emissions from 2.3 (1.9 - 2.8) Tg a[-1] to 3.1 (2.9 - 3.2) Tg a[-1], which would almost entirely explain the observed discrepancy between the mean GHGI23 and posterior emissions. The increase in manure management emissions is concentrated over the California Central Valley,

northern Iowa, and Sampson and Duplin Counties in North Carolina. California is home to more dairy cattle than any other state, Iowa is the largest pork-producing state, and Sampson and Duplin Counties are the two largest pork-producing counties in CONUS (USDA, 2019). We find no correlation between our inferred increase and dairy cattle or hog populations, which could reflect variability in manure management practices.

Posterior oil and gas emissions are 10.4 (10.1 - 10.7) Tg a[-1], a 12% increase from the GHGI23 estimate of 9.3 (8.1 - 10.6) Tg a[-1]. Lu et al. (2022) found much larger posterior emissions of 4.8 (3.1 - 4.9) Tg a[-1] for oil and 8.9 (8.0 - 9.8) Tg a[-1] for gas in 2017, and Lu et al. (2023) used the same inversion framework to find even larger total oil and gas emissions of 15.6 (12.8 - 17.1) Tg a[-1] for 2019 driven by increased emissions in the Anadarko, Marcellus, Barnett, and Haynesville Shales. Although we find good agreement on average with the basin-level emissions from Lu et al. (2023), we find much smaller emissions in

the Anadarko and Marcellus Shales, as shown in Fig. S3. This difference likely results in part from the use of lognormal prior errors in Lu et al. (2023). Compared to Lu et al. (2022, 2023), Worden et al. (2022) found smaller 2019 emissions in the United States for oil of $2.4 \pm 0.3$ Tg a[-1] and for gas of $7.9 \pm 0.9$ Tg a[-1], and Shen et al. (2022) found oil and gas emissions of $12.6 \pm 2.1$ Tg a[-1] from an inversion of TROPOMI data over 14 North American basins extrapolated to the national scale for May 2018 to February 2020. Both these emission estimates are within the uncertainty range of our posterior estimate. We

also find consistent basin-level results with Shen et al. (2022) as shown in Fig. S3. Emissions for all posterior basins but one are within 0.25 Tg a[-1] of Shen et al. (2022) and all but six are within 0.10 Tg a[-1]. In particular, we find agreement within error bars in the Haynesville, Barnett, and Anadarko Shales. Of the basins where posterior emissions exceed the 0.5 Tg a[-1] threshold defined by Shen et al. (2022) for successful quantification of basin emissions by TROPOMI, we find significant differences only in the Permian basin, where we find smaller emissions of 2.8 (2.8 - 2.9) Tg a[-1]. Our Permian estimate is

consistent within error bars with Lu et al. (2023) and with other recent studies when basin extent differences are accounted for (Zhang et al., 2020; Schneising et al., 2020; Liu et al., 2021; Varon et al., 2022; McNorton et al., 2022).

### 3.2 Landfill emissions

We consider in more detail the 51% increase in our posterior landfill emissions relative to the GHGI23. GHGI landfill estimates scale up the total emissions reported to the GHGRP to account for non-reporting landfills (EPA, 2023). The

GHGRP reporting requirements applied to 1297 landfills emitting more than 1 Gg a[-1] across the U.S. in 2019 (EPA GHGRP, 2019), over 500 of which had gas recovery systems (EPA LMOP, 2019). The GHGRP requires that landfills use two methods to report emissions. Facilities without gas collection use two approaches that rely on landfill attributes and a first-order decay model based on landfilled mass so that emissions peak the year after waste disposal. However, a survey of 128





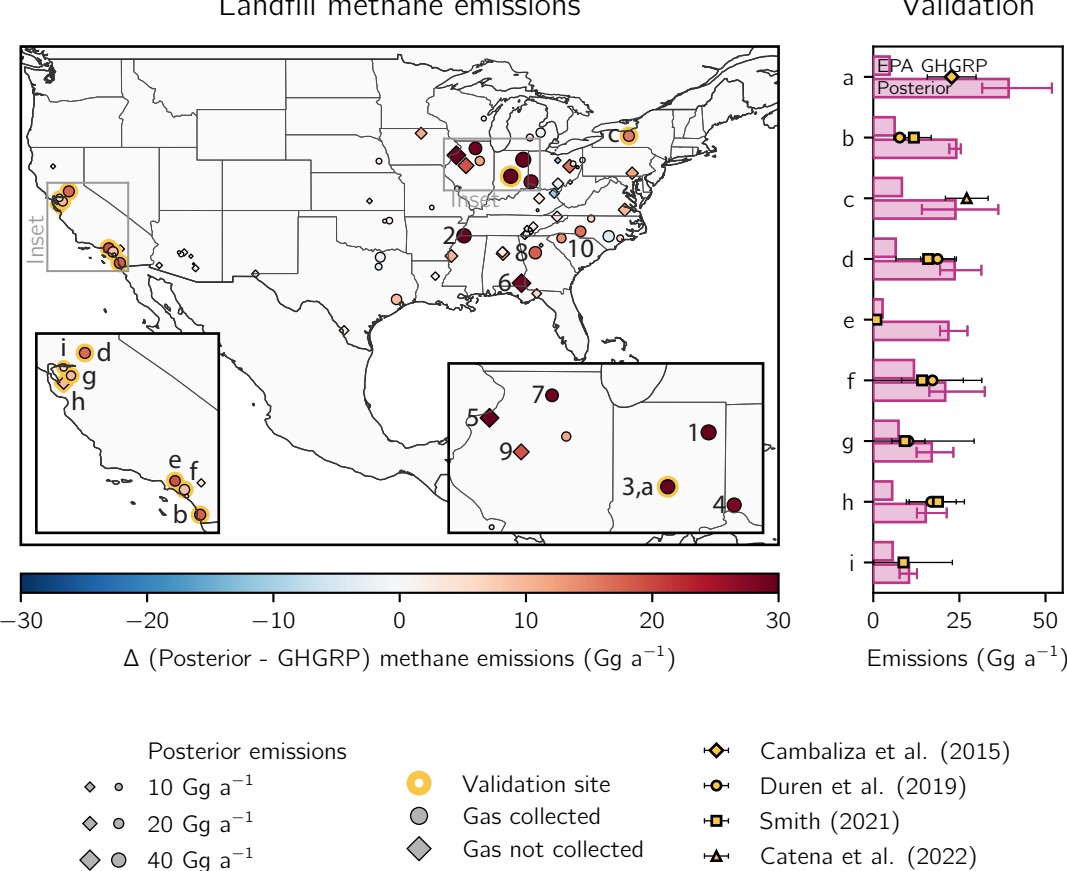

**Figure 5:** Methane emissions for 2019 from 73 individual landfills that report methane emissions of 2.5 Gg a$^{-1}$ or more to the EPA's Greenhouse Gas Reporting Program (GHGRP) for 2019 and for which our TROPOMI inversion provides site-specific information. The left panel shows the location of the landfills, with insets for parts of California (left) and Illinois and Indiana (right). Posterior emissions for each landfill are shown by the size of the marker. The colors show differences (Δ) between the posterior and GHGRP emissions for 2019, with red colors indicating posterior emissions larger than the reported value. Facilities that collect landfill gas are shown as circles, and others are shown as diamonds. The numbers (1 to 10) identify the top 10 methane-producing landfills listed in Table 3, and the letters (a to i) identify the nine validation sites listed in the right panel and outlined in gold. Validation sites are landfills with independent estimates from aircraft campaigns as listed in the legend. Cambaliza et al. (2015) based their estimates on data from 2011, Duren et al. (2019) on data from 2016 to 2018, Smith (2021) on data from 2019 to 2021, and Catena et al. (2022) on data from November 2021. The right panel shows GHGRP (top bars) and posterior (bottom bars) emissions for the validation sites, along with values reported from the aircraft campaigns. Sites are (a) South Side Landfill, (b) West Miramar Sanitary Landfill, (c) Seneca Meadows Landfill, (d) Kiefer Landfill, (e) Puente Hills Landfill, (f) Frank R. Bowerman Landfill, (g) Altamont Landfill, (h) Newby Island Landfill, and (i) Keller Canyon Landfill.

California landfills with gas recovery systems found that methane was produced at relatively constant rates over time (Spokas et al., 2015). Landfills with gas collection use one of these methods with recovered methane removed from the modelled emissions in addition to a back-calculation approach that estimates emissions as a function of recovered methane given an estimated collection efficiency based on cover and operation methods. A default efficiency of 0.75 is assumed if



cover information is unavailable (EPA, 2023). Both the model and back-calculation methods have high uncertainties and have not been field validated (NAS, 2018).

We compare our posterior landfill emissions to individual GHGRP facilities that reported more than 2.5 Gg a⁻¹ methane in 2019. Of these 531 landfills, we limit our analysis to the 87 0.25° × 0.3125° grid cells where TROPOMI provides an averaging kernel sensitivity greater than 0.20 and where landfills explain more than 50% of prior emissions so that we are confident of our ability to separate landfill emissions from other sources. We exclude 33 facilities in grid cells containing multiple landfills because we are unable to separate the individual contributions to total emissions. Figure 5 shows the

posterior emissions and corrections to the GHGRP for the remaining 73 facilities, Table 3 shows GHGRP and posterior information for the top 10 methane-producing landfills as ranked by posterior emissions, and Table S3 shows GHGRP and posterior information for all 73 facilities.

We validate our posterior landfill results by comparison to aircraft-derived estimates for nine facilities as shown in Fig. 5.
Cambaliza et al. (2015), Smith (2021), and Catena et al. (2022) used mass balance approaches to estimate emissions using observations from 2011, 2019 to 2021, and November 2021, respectively. Duren et al. (2019) used the integrated methane

**Table 3:** Top 10 methane-producing landfills in CONUS for 2019.

| Facility[1] | Location | Emissions (Gg a⁻¹) | | Gas capture efficiency | |
|---|---|---|---|---|---|
| | | GHGRP[2] | Posterior[3] | GHGRP[4] | Posterior[5] |
| 1. National Serv-All Landfill | Fort Wayne, Indiana | 3.4 | 44 (34 - 59) | 0.86 | 0.32 (0.26 - 0.37) |
| 2. South Shelby Landfill | Memphis, Tennessee | 4.1 | 41 (30 - 56) | 0.86 | 0.39 (0.31 - 0.46) |
| 3. South Side Landfill Inc. | Indianapolis, Indiana | 4.7 | 39 (32 - 52) | N/A | N/A |
| 4. Rumpke Sanitary Landfill | Cincinnati, Ohio | 10.1 | 39 (33 - 43) | 0.84 | 0.58 (0.55 - 0.61) |
| 5. Quad Cities Landfill Phase IV | Milan, Illinois | 3.7 | 35 (28 - 47) | N/A | N/A |
| 6. City of Dothan Sanitary Landfill | Dothan, Alabama | 5.8 | 35 (28 - 43) | N/A | N/A |
| 7. Rochelle Municipal Landfill | Rochelle, Illinois | 2.7 | 32 (25 - 39) | 0.76 | 0.22 (0.18 - 0.26) |
| 8. Seminole Road MSW Landfill | Ellenwood, Georgia | 12.3 | 30 (25 - 36) | 0.18 | 0.08 (0.07 - 0.1) |
| 9. Caterpillar Inc.-Mapleton | Mapleton, Illinois | 6.4 | 25 (23 - 29) | N/A | N/A |
| 10. Sampson County Disposal, LLC | Roseboro, North Carolina | 29.2 | 25 (23 - 29) | 0.37 | 0.41 (0.38 - 0.44) |

[1]The top 10 landfills with the largest posterior methane emissions from the TROPOMI inversion for 2019. Numbers correspond to the
labels in Fig. 5.
[2]Emissions reported by individual landfills to the EPA GHGRP for 2019 in gigagrams per year.
[3]Posterior emissions from the inversion of TROPOMI observations in gigagrams per year. Posterior emissions are allocated to individual facilities as described in Sects. 2.8 and 3.2. Values in parentheses represent the range from the eight-member inversion ensemble.
[4]For facilities that capture landfill gas, the recovery efficiency as calculated from emissions and recovered methane reported by individual
landfills to the EPA LMOP. Facilities that do not capture landfill gas are listed as N/A.
[5]The posterior recovery efficiency as calculated from posterior emissions and the recovered methane reported by individual landfills to the EPA LMOP.





enhancement method with data from 2016 to 2018. We find agreement within error bounds at the Seneca Meadows Landfill in New York (landfill c in Fig. 5; Catena et al., 2022) and at the Kiefer (d), Frank R. Bowerman (f), Altamont (g), Newby

Island (h), and Keller Canyon (i) Landfills in California (Smith, 2021; Duren et al., 2019). We find much larger emissions than previous studies at the South Side Landfill (a) in Indiana (Cambaliza et al., 2015) and at the West Miramar Sanitary (b) and Puente Hills (e) Landfills in California (Smith, 2021; Duren et al., 2019). The discrepancy at the South Side Landfill could reflect changed emissions since 2011, including the construction of a large landfill gas recovery facility beginning in June 2019 (EPA LMOP, 2019). Methane concentrations of 8662 ppm were recorded at a leak at the West Miramar Sanitary

Landfill in November 2019 (San Diego Air Pollution Control District, 2019), suggesting that estimates from other years may not be representative of 2019 emissions. The Puente Hills Landfill closed in 2013 but was previously one of the largest landfills in CONUS (EPA GHGRP, 2019). Our landfill attribution approach, which relies on a prior estimate from 2012, may therefore misallocate emissions to the Puente Hills Landfill instead of to co-located oil and gas operations.

We find mean facility emissions of 13 Gg a$^{-1}$ compared to the GHGRP mean of 7.2 Gg a$^{-1}$ for the 73 landfills considered here, with a median 77% increase in reported emissions. As reflected in Table 3, we find no correlation ($R^2 = 0.00$) between GHGRP emissions and our posterior estimates, which does not improve when we consider only facilities that do or do not capture landfill gas. This implies that the bottom-up approaches used for emissions estimation have little predictability.

For the 38 facilities that recover gas, we use captured methane emissions reported to the EPA Landfill Methane Outreach Program (LMOP) in 2019 together with posterior and GHGRP emissions to calculate a posterior and reported recovery efficiency, respectively. We find a low correlation ($R^2 = 0.17$) between the efficiencies that does not depend on facility size but improves slightly for facilities constructed within the last decade ($R^2 = 0.31$). The average posterior recovery efficiency of 0.50 (0.33 - 0.54) is much smaller than the GHGRP mean of 0.61, and both are much smaller than the 0.75 default (EPA,

2023). Across the six landfill gas facilities at the top 10 methane-producing landfills, we find a mean posterior recovery efficiency of 0.33 that is half the GHGRP value of 0.65. Indeed, four of the six facilities report methane emission and recovery values consistent with efficiencies larger than the 0.75 default. We find a similar but still lower efficiency at the Seminole Road MSW Landfill (landfill 8) and a marginally higher recovery efficiency only at Sampson County Disposal, LLC (10).


We consider in detail the 34 facilities for which posterior emissions show a significant 50% difference from the GHGRP. We find larger emissions for 29 of these facilities, with the largest discrepancies occurring in nine of the top 10 methane-producing landfills. Three of these nine facilities experienced significant operational changes in the last decade. The South Shelby (landfill 2 in Fig. 5) and South Side (3) Landfills constructed large landfill gas facilities in 2019 (EPA LMOP, 2019;

Russell, 2019), suggesting that emissions from gas infrastructure development may be large. The City of Dothan Sanitary Landfill (6) has been full since 2014, when it stopped accepting most trash (Wise, 2019). Reported emissions peaked at 7.4





Gg a$^{-1}$ that year (EPA GHGRP, 2019), a value almost five times smaller than our posterior emissions, suggesting that the first order decay model is inadequate to reproduce methane emissions over time. We also find a record of air quality and landfill standard violations at these 34 facilities. At the West Miramar Sanitary Landfill (b), a leak emitting 8662 ppm

methane was recorded in November 2019 (San Diego Air Pollution Control District, 2019). The Sussex County Landfill in Virginia was fined USD 99000 in 2016 for failing to address cracks in the landfill cover (Vera, 2016). Lastly, the Newby Island Landfill (h), received 30 violation notices from 2014 to 2020, including for gas collection system shutdowns (Bay Area Air Quality Management District, 2022).

There are five facilities for which our posterior emissions are significantly smaller than the 2019 GHGRP by 50%. Three report large decreases in estimated methane emissions from 2019 to 2020 that result from changed methodology (EPA GHGRP, 2019). The updated estimates are consistent with our posterior emissions within error estimates in two cases and within 30% of our posterior emissions in the third case.

**3.3 State emissions**

The EPA recently began disaggregating the GHGI by state. The EPA uses the same methods to calculate state emissions as in the national inventory so that the total emissions are the same in both estimates. We use the most recent state inventories available as published with the GHGI22. The GHGI23 national emission estimate for 2019 increases only 2% from the GHGI22 value, suggesting that the unreleased GHGI23 state emissions should be similar to the GHGI22 estimates. State estimates are developed without reference to greenhouse gas inventories prepared by state governments, which may result in

discrepancies in sectoral or total values due to different methods or accounting (EPA, 2022b). In addition to the GHGI22 state estimates, the EPA provides references to the independent inventories of 24 states and Washington, D.C. (EPA, 2023). Of these, we find that eight produce a methane emission estimate separate from their inventory of total $CO_2$-equivalent greenhouse gases.

We partition our anthropogenic gridded posterior emission estimates, excluding offshore emissions, to each of the 48 states in CONUS as described in Sect. 2.8 and compare the results to the GHGI22 state estimates and to inventories prepared by state governments. Figure 6 shows the results for the 29 states responsible for 90% of posterior CONUS anthropogenic emissions excluding offshore emissions and ordered by posterior emissions, and Table S1 shows the full results for all 48 CONUS states. TROPOMI provides a strong constraint at this resolution, with most state averaging kernel sensitivities larger

than 0.5. Our state emissions are on average 10% larger than the GHGI22 estimates and 34% larger in the top 10 methane-emitting states, which produce 55% of CONUS posterior emissions. Oil and gas emissions on average generate 37% of posterior emissions and 46% of the observed increase relative to the GHGI22 in these 10 states. In Texas, New Mexico, Louisiana, and Oklahoma, the oil and gas sector explains more than 60% of posterior emissions, with emissions concentrated in the Permian Basin, the Haynesville Shale, and the Anadarko Shale. The addition of basin-specific information in the

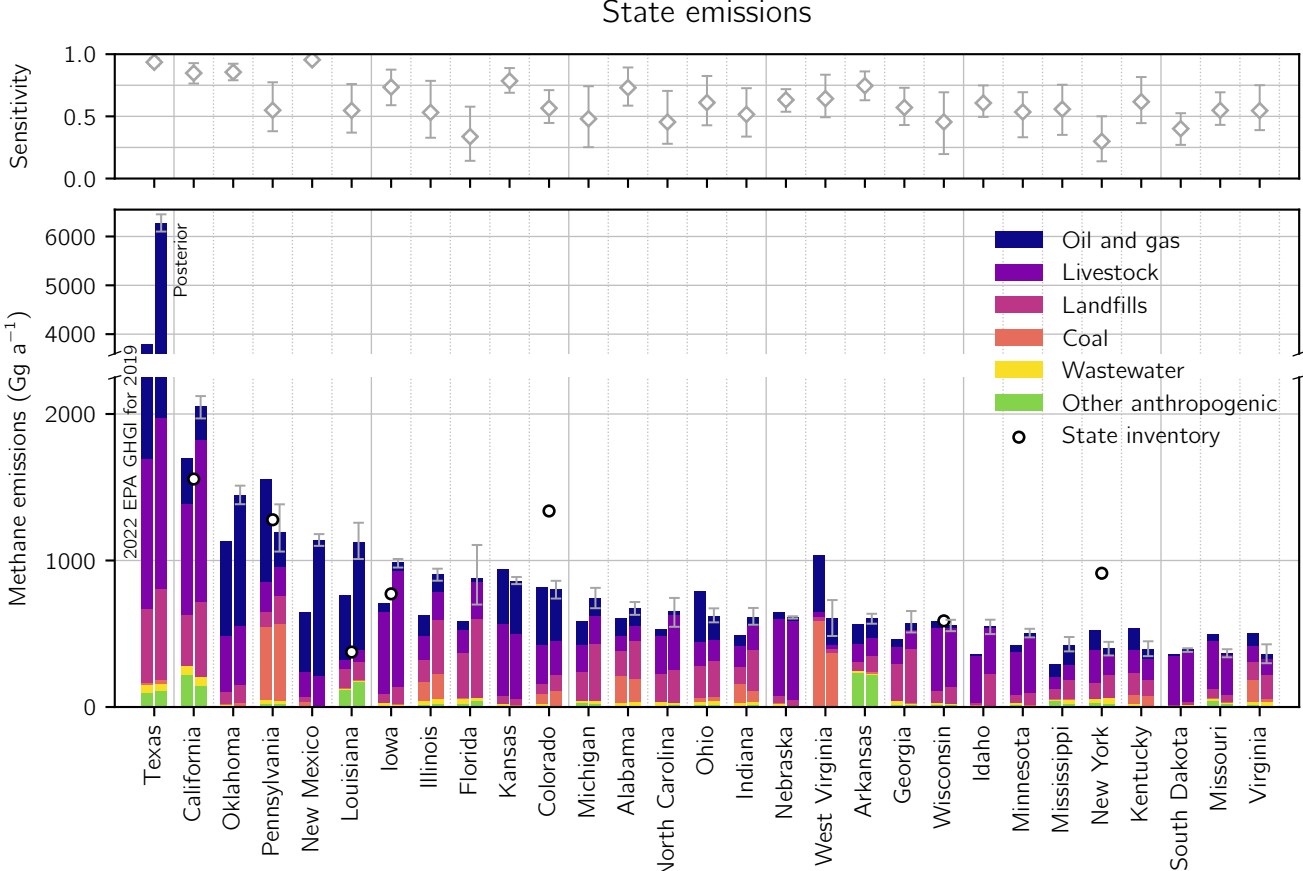

**Figure 6:** Anthropogenic methane emissions in 2019 for the 29 states responsible for 90% of U.S. anthropogenic posterior emissions. The bottom panel shows 2022 EPA GHGI state estimates for 2019 (left bar) and our posterior estimates from the inversion of TROPOMI data (right bar) divided by sector. States are listed from largest to smallest posterior emissions. The information content from the TROPOMI data as defined by the reduced-form averaging kernel sensitivities (the diagonal elements of the reduced-form averaging kernel matrix; Sect. 2.8) is shown in the top panel. Values of 1 indicate full sensitivity to TROPOMI, while values of 0 indicate no sensitivity. The error bars give the spread from the eight-member inversion ensemble. Also shown are emissions estimates from independent state inventories referenced by EPA (2022).

GHGI23 may improve the state-level distribution of oil and gas emissions (EPA, 2023). Livestock and landfills also play a significant role in these states. Emissions in California and Iowa are dominated by the livestock sector, with much of the observed increase relative to the GHGI22 attributed to manure management emissions (Sect. 3.1). Landfills account for 41% of posterior emissions in Illinois and 62% in Florida. Indeed, three of the ten largest landfills as reported to the GHGRP in 2019 are in Florida (EPA GHGRP, 2019). Consistent with our sectoral analysis, the largest posterior emission decreases relative to the GHGI22 are found in coal-producing states, including West Virginia and Pennsylvania. While we find a large decrease compared to the GHGI in Pennsylvania, we cannot confidently attribute the difference to a specific sector due to co-location of oil, gas, and coal facilities at the resolution of our inversion.





We consider in more detail Texas and California, which are responsible for 21% and 7% of posterior CONUS anthropogenic emissions, respectively. Our posterior estimate for Texas is 6.3 (6.1 - 6.5) Tg a$^{-1}$, a 66% increase from the GHGI22 estimate of 3.8 Tg a$^{-1}$. This increase is attributed almost entirely to the oil and gas sector, which accounts for 69% of posterior

emissions compared to 55% in the GHGI22. The Permian basin alone explains almost 40% of Texas' posterior emissions. In California, we find posterior emissions of 2.1 (2.0 - 2.1) Tg a$^{-1}$, 53% of which occur in the San Joaquin Valley air basin. Our posterior emissions increase 21% from the GHGI22 estimate of 1.7 Tg a$^{-1}$ and 32% from an independent estimate produced by CARB of 1.6 Tg a$^{-1}$ (CARB, 2022). Our posterior estimate is smaller than but consistent within error bars with a value of 2.4 ± 0.5 Tg a$^{-1}$ found by an inversion of in situ observations in California from June 2013 to May 2014 (Jeong et al., 2016).

We find in general good agreement with the sectoral partitioning in the GHGI22, the CARB inventory, and Jeong et al. (2016). Livestock explain 54% of emissions in our posterior estimate, 45% in the GHGI22, 54% in the CARB inventory, and 54% in Jeong et al. (2016), while landfills explain 25%, 20%, 21%, and 19% of emissions, respectively. We find slightly smaller relative contributions from oil and gas, which accounts for 11% of emissions in our posterior estimate compared to 18%, 17%, and 18% in the GHGI22, the CARB inventory, and Jeong et al. (2016), respectively. This partitioning differs

from that found in an inversion of the 2010 CalNex aircraft campaign observations, where 30% of emissions were attributed to livestock, 38% to landfills, and 22% to oil and gas based on the sectoral distribution of the EDGAR v4.2 methane emission inventory (Wecht et al., 2014b).

We also compare our posterior emissions to independent state greenhouse gas inventories from Pennsylvania, Louisiana,

Iowa, and Colorado referenced by EPA (2023), where we have a strong constraint from the inversion (state averaging kernel sensitivity greater than 0.5). Our posterior agrees with the Pennsylvania estimate (Pennsylvania DEP, 2022), but we find a source shift from fossil fuels (from 76% in the inventory to 63% in our work) to landfills (from 3% in the inventory to 16% in our work). We find that Louisiana's state inventory (Dismukes, 2021) is too low due to underestimated oil and gas emissions, while Iowa's (Iowa DNR, 2020) is too low due to underestimated livestock emissions, particularly from manure

management (Sect. 3.1). Colorado's state inventory (Taylor, 2021) is 65% larger than our posterior estimate due to oil and gas emissions that are more than twice as large.

### 3.4 Urban area emissions

Urban areas are home to 81% of the U.S. population (U.S. Census Bureau, 2010) and are major sources of greenhouse gas emissions, including methane (Gurney et al., 2015; Hopkins et al., 2016). As urban populations grow (Seto et al., 2012),

these emissions are likely to increase. Cities are well positioned to address methane emissions through waste-reduction initiatives, leak-detection programs, and strategic contracts with landfill operators and gas utilities. Regulation by air pollution control districts can also aid urban emissions reduction efforts (Hopkins et al., 2016). C40, a performance-based coalition of over 100 mayors dedicated to climate change mitigation, recommends that cities target a 50% reduction in methane emissions by 2030 (C40, 2022b). Numerous cities, including New York City, Los Angeles, and Philadelphia, are



working toward these reductions through zero-waste programs (C40, 2022a). The U.S. Methane Emissions Reduction Action
       Plan intends to work with local governments to set up methane monitoring systems to identify and publicize information
       about municipal gas distribution leaks. The plan also challenges members of the U.S. Climate Mayors to prioritize pipeline
       abandonment or replacement (The White House, 2021).

We calculate posterior emissions for 95 urban areas across CONUS with 2010 populations over 1 million and averaging
       kernel sensitivities from our inversion greater than 0.2, providing the first comprehensive national analysis of urban methane
       emissions. Quantification of urban emissions depends significantly on the definition of city extent due to the presence of
       large emitters such as landfills on the urban periphery (e.g., Balashov et al., 2020; Plant et al., 2022). We follow Plant et al.
       (2022) and others in using the U.S. Census Topographically Integrated Geographic Encoding and Referencing system
(TIGER)/Line Urban Areas to standardize the definition across CONUS (U.S. Census Bureau, 2017). These urban areas are
       responsible for almost a quarter of the GHGI23 emissions spatially allocated using the gridded inventory from Maasakkers et
       al. (2016). The gridded inventory does not include post-meter emissions introduced in later versions of the GHGI, which we
       distribute by population for this analysis. In an average city, the gridded GHGI emissions originate from landfills (40%), gas
       distribution (9%, including 4% from post-meter emissions), wastewater (6%), and other sources that are not specific to urban
areas such as livestock and oil and gas production and transmission (45%).

       Anthropogenic posterior emissions in these 95 urban areas total 6.0 (5.4 - 6.7) Tg a$^{-1}$, 38 (24 - 54) % larger than the gridded
       GHGI23 value of 4.3 Tg a$^{-1}$. Individual urban area emissions, listed in Table S2, increase by an average of 39 (27 - 52) %.
       These increases are much larger than the 13% increase we find in total CONUS anthropogenic emissions relative to the
GHGI23. We are unable to attribute the increased emissions to individual sectors due to source co-location within urban
       areas at the 0.25° × 0.3125° resolution of our inversion. However, given that landfills account for 40% of gridded GHGI23
       emissions in an average urban area and increase 51% relative to the GHGI23, it is likely that they are responsible for a large
       fraction of the observed discrepancy. It is also likely that gas emissions, which represent less than 20% of gridded GHGI23
       emissions in an average urban area but explain between 32% and 100% of methane emissions in many cities based on field
measurements of methane-ethane ratios (Plant et al., 2019; Floerchinger et al., 2021; Sargent et al., 2021), are significantly
       underestimated. Finally, recent studies have shown large underestimates of methane emissions from wastewater treatment in
       the GHGI (Moore et al., 2023; Song et al., 2023) and over urban areas (de Foy et al., 2023), but increasing wastewater
       emissions accordingly only accounts for 2% of our observed discrepancy. City-specific variability prevents further
       attribution of urban emissions. Indeed, we find no correlation between the posterior emission increase and urban area
population, population change from 2000 to 2010, population density, or surface area.

       Figure 7 shows results for the top 10 methane-producing urban areas as ranked by posterior emissions from landfills, gas
       distribution, and wastewater. These 10 regions explain 35 (34 - 36) % of anthropogenic posterior emissions across the 95





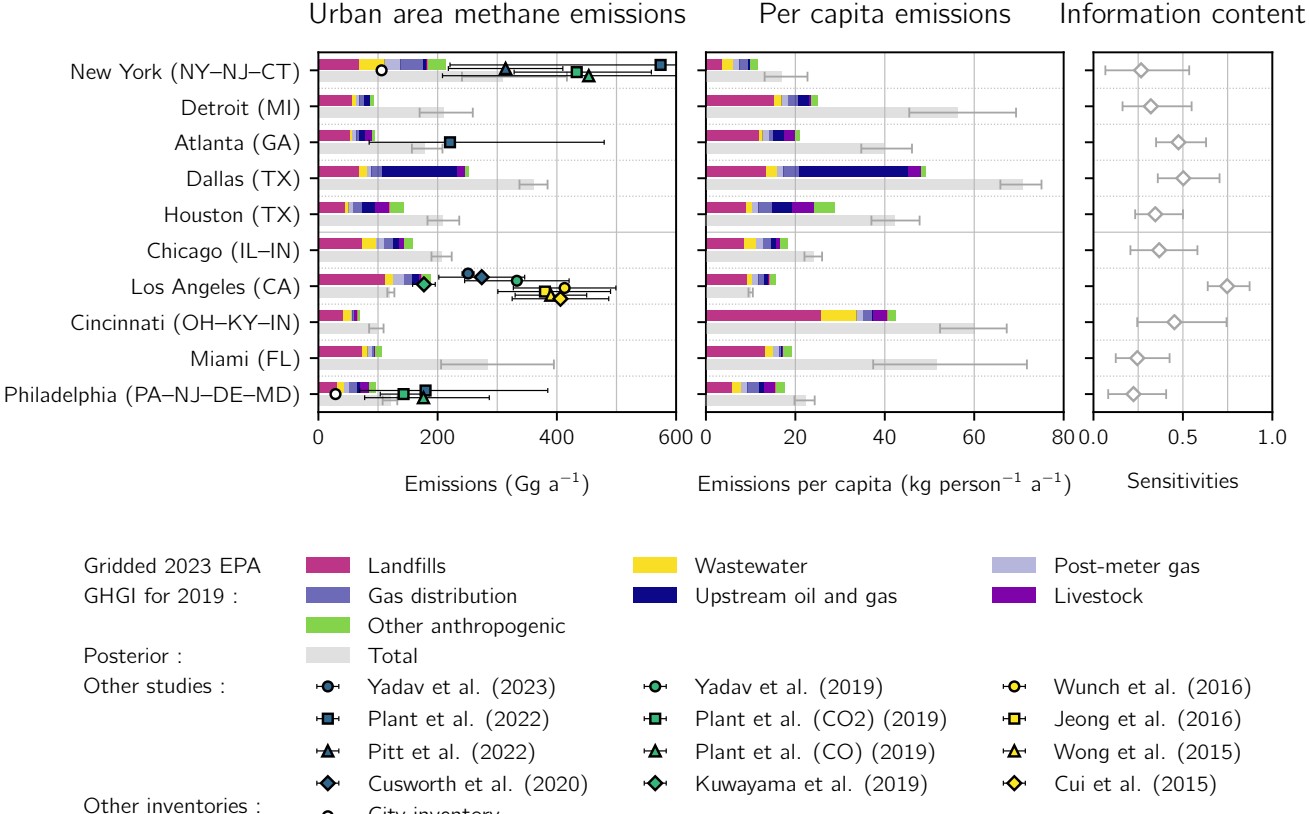

**Figure 7:** Anthropogenic methane emissions for the largest 10 methane-producing urban areas in the contiguous United States (CONUS) for 2019 as identified by the inversion of TROPOMI data. Urban area extents are given by the U.S. Census Bureau TIGER/Line files (U.S. Census, 2010). The top bars show prior anthropogenic sectoral emissions from the 2023 EPA GHGI for 2019 spatially allocated following Maasakkers et al. (2016) with post-meter emissions allocated by population. The bottom bar shows posterior emissions from the TROPOMI inversion for 2019. We do not resolve posterior sectoral emissions estimates due to source colocation within urban areas at the scale of the inversion. Total emissions (left panel), per capita emissions (center panel), and averaging kernel sensitivities (right panel) are shown for each urban area. Error bars represent the spread of the eight-member inversion ensemble. Also shown are independent urban emissions estimates.

urban areas considered here. We find a mean increase relative to gridded GHGI23 emissions of 58 (37 - 84) %. We also compare our posterior emissions to municipal inventories from New York City and Philadelphia, the only available bottom-up urban methane emission estimates. Our emissions are more than twice as large as these inventories, but this likely results from our consideration of broader urban areas.

Figure 7 also compares our results to 12 top-down studies published since 2015. Most of these focused on New York City or Los Angeles. Almost all the studies used larger definitions of urban area extent, with only Pitt et al. (2022) and Plant et al. (2022) using the U.S. Census designation. Most used aircraft or tower observations to infer emissions by inverting a CTM (Cui et al., 2015; Jeong et al., 2016; Cusworth et al., 2020; Pitt et al., 2022; Yadav et al., 2019, 2023). Kuwayama et al.



(2019) used a mass balance approach, while others used observed methane to $CO_2$ or CO ratios together with bottom-up inventories of these gases (Wong et al., 2015; Wunch et al., 2016; Plant et al., 2019). Plant et al. (2022) used the same approach with TROPOMI methane to CO emissions.


We find in general lower but statistically consistent emissions compared to these studies. Our smaller estimates likely result from our restrictive definition of urban area extent. The only study that used aircraft data to estimate emissions within a U.S. Census Urban Area found $314 \pm 96$ Gg a$^{-1}$ in New York City (Pitt et al., 2022), which is very similar to our estimate of 309 (241 - 417) Gg a$^{-1}$. Plant et al. (2022) used U.S. Census Urban Areas but relied on TROPOMI methane to CO ratios. They

found slightly larger emissions in Atlanta and Philadelphia and much larger emissions in New York City, but their error bars spanned ranges almost twice as large as the derived emissions, limiting the utility of the comparison. Plant et al. (2019) found larger emissions in New York City and Philadelphia but used larger definitions of urban areas and produced similarly wide error ranges.

We find much lower emissions than these studies only in Los Angeles, a difference that decreases but remains significant when we use the same extent as these studies. We attribute much of the discrepancy to decreasing emissions over time. Methane emissions from the Puente Hills Landfill, previously one of the largest landfills in CONUS, decreased following its closure in 2013 (Yadav et al., 2019). This change is not fully reflected in the estimates of Cui et al. (2015), Wong et al. (2015), or Wunch et al. (2016). Yadav et al. (2023) found that Los Angeles emissions decreased an additional 7% from

January 2015 to May 2020. However, their posterior estimate of $251 \pm 5$ Gg a$^{-1}$ for 2019 is still larger than our value of 179 (171 - 193) Gg a$^{-1}$.

## 4 Conclusions

We used TROPOMI atmospheric methane column observations for 2019 to optimize methane emissions at $0.25° \times 0.3125°$ resolution over North America with a focus on the contiguous U.S. (CONUS). The high resolution of our inversion allowed

us to quantify emissions from individual landfills, states, and urban areas. We compared our results to the 2023 EPA Greenhouse Gas Emissions Inventory (GHGI) for 2019; to new EPA state-level inventories for 2019 published most recently with the 2022 GHGI; to emissions reported by individual landfills to the EPA Greenhouse Gas Reporting Program (GHGRP); and to other estimates from states and cities. We find large upward corrections to the GHGI at all scales, which may present a challenge for U.S. climate policies and goals, many of which target significant reductions in methane

emissions.

We optimized methane emissions using an analytical inversion of TROPOMI methane observations with the GEOS-Chem chemical transport model run at $0.25° \times 0.3125°$ resolution. The inverse solution, or posterior emission estimate, was



obtained through a reduced-rank approximation of the analytical minimum of a Bayesian cost function regularized by a prior
emission estimate from a gridded version of the GHGI. The analytical solution characterizes the error and information
content of the posterior emissions and supported the generation of an eight-member inversion ensemble. We constructed the
Jacobian matrix required for the high-resolution, continent-scale analytical solution by iterative approximation using the
emissions patterns best informed by the prior emission estimate and the observations. This approach decreases the
computational cost of our inversion by an order of magnitude compared to conventional analytical methods while optimally
preserving its information content.

We find posterior anthropogenic methane emissions of 30.9 (30.0 - 31.8) Tg a$^{-1}$ in CONUS, where the range is given by the
inversion ensemble. This is a 13% increase from the 2023 GHGI estimate of 27.3 (25.1 - 30.6) Tg a$^{-1}$, where the range is
given by the 95% confidence interval. Emissions for landfills, oil and gas, and livestock explain 89% of posterior CONUS
emissions and each of these sectors' emissions increase by at least 10% relative to the GHGI. We find a significant decrease
compared to the 2023 GHGI only for coal emissions. These increases present a challenge to goals set by the U.S.
government to decrease methane emissions from landfills by 30% relative to 2015 levels by 2025 and to regulation in
development that aims to reduce oil and gas methane emissions by 30% from 2020 to 2030.

Most of the total increase from the 2023 GHGI to the posterior emissions is attributed to a 51% increase in landfill
emissions. We compare our optimized emissions for 73 individual landfills to those reported to the GHGRP and find a
median 77% increase in emissions relative to reported values. We attribute the underestimated GHGI and GHGRP landfill
emissions to standard inventory methods that (1) assume too-high recovery efficiencies at facilities that collect landfill gas
and (2) inadequately account for anomalous operating events such as gas leaks or the construction of new landfill gas
facilities.

We took advantage of the high resolution of our inversion to quantify emissions for each of the 48 states in CONUS and
compare to the newly available state emission inventories published most recently with the 2022 GHGI. We find a 10%
average increase with a 34% average increase in the top 10 methane-emitting states. Much of the discrepancy in these 10
states is attributed to increased oil and gas emissions, though livestock and landfills also play significant roles. Texas and
California, the two largest methane-producing states, respectively emit 21% and 7% of total CONUS anthropogenic
emissions in our posterior estimate. Emissions in Texas increase by 66% relative to the 2022 GHGI almost entirely due to
the oil and gas sector. Operations in the Permian basin alone explain almost 40% of all posterior emissions in the state. In
California, we find a 21% increase from the 2022 GHGI and a 32% increase from an independent inventory prepared by the
California Air Resources Board (CARB). Our sectoral partitioning for California is consistent with both inventories,
including 54% of emissions from livestock, 25% from landfills, and 11% from oil and gas.

We also provide a first national analysis of urban methane emissions by calculating emissions for 95 urban areas across
CONUS. We find total emissions of 6.0 (5.4 - 6.7) Tg a$^{-1}$ across these urban areas, representing a fifth of posterior
anthropogenic emissions in CONUS and a 38 (24 - 54) % increase from the gridded 2023 GHGI value of 4.3 Tg a$^{-1}$. Urban
emissions increase on average by 39 (27 - 52) % compared to the GHGI. We attribute the observed discrepancy to
underestimated landfill and gas emissions. Our urban emission estimates are in general consistent with previous top-down
studies except for Los Angeles.

**Code availability**

The GEOS-Chem code is available at https://doi.org/10.5281/zenodo.3676008, and the description of the model is available
at geos-chem.org. The code to solve and analyze the inversion is at https://github.com/hannahnesser/TROPOMI_inversion.

**Data availability**

The TROPOMI v14 data are available from SRON at https://ftp.sron.nl/open-access-data-2/TROPOMI/tropomi/ch4/14_14_Lorente_et_al_2020_AMTD/ (last access: 19 March 2021). The GLOBALVIEWplus CH$_4$
ObsPack v3.0 database is available from NOAA's Global Monitoring Laboratory at http://dx.doi.org/10.15138/G3CW4Q.
The prior and observational inputs for the inversion and the posterior emissions and averaging kernel sensitivities are
available at https://github.com/hannahnesser/TROPOMI_inversion. Additional data related to this paper may be requested
from the authors.

**Author contribution**

HN and DJJ designed the study. HN conducted the inversion with contributions from ZC, ZQ, MPS, and MW. SM and AAB
provided the high-performance ensemble of WetCHARTS v1.3.1 and supported wetland analysis. JDM and AL provided
guidance on the TROPOMI data. JDM, AL, XL, LS, JW, RNS, and CAR discussed the results. HN and DJJ wrote the paper
with input from all authors.

**Competing interests**

The contact author has declared that none of the authors has any competing interests.



## Acknowledgments

This work was supported by the NASA Carbon Monitoring System (CMS), ExxonMobil Technology and Engineering Company, and the Harvard Climate Change Solutions Fund. Part of this work was carried out at the Jet Propulsion Laboratory, California Institute of Technology, under a contract with the National Aeronautics and Space Administration
(NASA). We thank Bryan Mignone, Felipe J. Cardoso-Saldaña, Robert Stowe, Lauren Aepli, and Melissa Weitz for helpful discussions.

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
