# Peer review of "High-resolution U.S. methane emissions inferred from an inversion of 2019 TROPOMI satellite data: contributions from individual states, urban areas, and landfills"

_EGUsphere, 2023_

## Author Comment (AC1)

We thank both reviewers for their comments on our manuscript, which provided valuable insights. Please see below our responses to the reviewers's comments. The comments are italicized, with our responses in standard text. Blue font corresponds to manuscript text, with changes underlined or struck through. We have also attached a tracked-changes version together with the full (all changes accepted) version.

In addition to the responses to the reviewers, we made several modifications to the text that leave the results unchanged:

1. We updated our comparison to the GHGI state estimates to reflect the 2023 inventory. Corresponding quantitative edits were made throughout the manuscript to reflect the updated results; the qualitative results remain the same.
2. We removed three industrial landfills from our list of landfills.
3. We clarified the emission reduction goals set by the U.S. government.

**Reviewer 1**

*Nesser et al., present detailed results for methane emissions in the contiguous U.S. as derived via inverse modelling using satellite observations of methane columns. They present new and very interesting results appropriate for EGUsphere and the paper is very well written. I strongly recommend publication in EGUsphere after the issues listed below have been carefully addressed by the authors.*

*Most of my comments relate to minor issues (see below) but there is one major aspect:*

*Whereas an ensemble of inversions has been used to obtain reliable results (which is very good) only one satellite data product has been used. Taking into account the high (also political) relevance of the results - the authors present very detailed emission results for various sectors and down to quite high spatial resolution (state, urban and even facility level) - I wonder if this is really appropriate. I have doubts in particular because an initial (i.e., quite old) satellite data product has been used which has known bias issues. Several of these bias issues are listed in the manuscript (line 209 following). It is written in the paper that because of the bias issues the authors could not use the data as they are but had to correct and filter them. For example, they had to remove data with too low methane values (e.g., data below 1700 ppb as these data were considered unrealistic). Bias corrections done by users are not uncommon but should be avoided if better data exist. On line 355 the authors write that they are aware of better data: „We also compare the TROPOMI v14 data used here to the most recent data (v19), which has improved bias corrections and performance compared to GOSAT in North America (Balasus et al., 2023). We find no correlation (R2 = 0.03) between our posterior scaling factors and the mean (v14 - v19) difference, suggesting that biases in the v14 data do not influence our posterior emissions." While this is good to know, this is by far not sufficient. The high spatial resolution methane emission information is contained in small scale methane gradients and even small differences of the methane data can have a large impact on the derived emissions. I therefore strongly recommend repeating the analysis using the latest (v19 or higher) product.*
Thank you for your valuable comment. While we agree that such an improvement would be beneficial and an important component of future work, redoing this analysis with the updated data would require redoing all ~2000 of the 0.25° × 0.3125° perturbation simulations, representing over 100 thousand hours of computation time. The time and computational resources required are beyond the scope of this manuscript as the lead author has started a new position.

In recognition of the importance of avoiding the propagation of potential systematic biases in the TROPOMI v14 data, we expand the analysis mentioned by the reviewer on line 355 as follows:

We also compare the TROPOMI v14 data used here to the most recent data (v19), which has improved bias corrections and performance compared to GOSAT in North America (Balasus et al., 2023). We define the grid cells containing observations sensitive to the optimized emissions by calculating the row-wise sum of the Jacobian matrix weighted by the prior and observing system error standard deviations, limited to emission grid cells with averaging kernel sensitivities greater than 0.05. Of the 95% of observation grid cells most sensitive to the optimized emissions, only 14% have an average mean-bias corrected (v14 – v19) difference greater than 5 ppb and less than 2% have a difference greater than 10 ppb. We similarly define

the observation grid cells that influence the optimized emission grid cells. We find that the average mean-bias corrected (v14 – v19) difference for the observational grid cells influenced by optimized grid cells is -0.05 ppb with a standard deviation of 0.1 ppb, indicating that there is little bias in the observations that influence any single grid cell. We finally find no correlation ($R^2 = 0.03$) between our posterior scaling factors and the mean (v14 - v19) difference, suggesting that biases in the v14 data do not influence our posterior emissions.

We will also point to our in-depth comparison to other satellite and aircraft emission quantifications at every point possible (see for example our comparison to GOSAT inversions in Figure 4, our comparison to aircraft studies throughout Section 3.1, our comparison to aircraft studies in Figure 5, our comparison to an in situ study in California, and our comparison to aircraft, in situ, and satellite studies in Figure 7).

*If resources permit, I recommend to also use an ensemble approach for the satellite data (even if only 2-3 products can be used) given the high relevance of the results and that these satellite data are the key input data for the inversion. This would give additional strong confidence that the quite large reported differences compared to the EPA inventory are a reliable finding and do not significantly depend on the peculiarities of a given satellite data product. In this context I would expect that the ensemble-based results will have larger (and likely more realistic) error bars than the (surprisingly small) error bars as shown in, e.g., Fig. 4. In this case I suggest to include also the TROPOMI WFMD v1.8 XCH4 data product described in Schneising et al., 2023. In that publication several comparisons with the operational TROPOMI product are shown (note that this operational RemoTeC algorithm product is very similar as the scientific RemoTeC product used here) indicating higher accuracy of the WFMD product due to much reduced bias (e.g., much reduced "striping errors" etc.). Such a satellite data ensemble approach would strengthen the manuscript even further, which is relevant here given the importance of the paper and the level of detail reported.*

Thank you for your valuable comment. As described above, the time and computational resources required to address this comment are beyond the scope of this manuscript as the lead author has begun a new position. As described in the response above, adding a second satellite product would require redoing all ~2000 of the 0.25° × 0.3125° perturbation simulations, representing over 100 thousand of hours of computation time. We instead compare to other satellite and aircraft emission quantifications at every point possible as described above and in the manuscript.

We also note that the small error bars in Figure 4 are due in large part to the quality-control applied to the inversion ensemble. As a result, it is not necessarily the case that the addition of a second product would increase the error bar magnitude. Indeed, we include two sets of observational data (one with a mean bias correction to the model – observation difference, and one with a latitudinal bias correction to the model – observation difference) and find small error bars.

*Minor aspects:*

*Line 61: Not sure if the "3% success rate" is valid for the real data used here as the cited (quite old) document (Hasekamp et al., 2019) covers algorithm related theoretical aspects not based on multi-year statistics.*

Thank you for your comment. Alba Lorente (co-author and former member of the TROPOMI CH4 SRON team) and Mari C. Martinez-Velarte (current member of the TROPOMI CH4 SRON team) confirmed in a private correspondence that the average percentage of pixels that converge after applying the post-filter is 3.5%. As a result, we opt to keep the Hasekamp et al. citation.

*Sect. 2: The time dependence of the emissions is prescribed and not optimized, or? If yes, then this needs to be explicitly mentioned somewhere.*

Thank you for your comment. We have clarified throughout the manuscript that we optimize annual-average methane emissions.

Abstract:

We quantify 2019 annual mean methane emissions in the contiguous U.S. (CONUS) at 0.25° × 0.3125° resolution by inverse analysis of atmospheric methane columns measured by the Tropospheric Monitoring Instrument (TROPOMI).

Introduction:
Here we use the reduced-rank method of Nesser et al. (2021) in an analytical inversion of 2019 TROPOMI observations to quantify annual mean emissions at 0.25° × 0.3125° resolution over North America using national emission inventories reported by the U.S., Mexico, and Canada to the UNFCCC as prior estimates.

Section 2 (Data and methods):
The $m = 2919358$ TROPOMI observations are fit to simulated GEOS-Chem concentrations to optimize annual mean methane emissions for 2019 at the 0.25° × 0.3125° GEOS-Chem resolution

Section 2.2 (Prior estimates and errors):
Figure 1 shows the annual-average prior emission estimates for different sectors.

Figure 1 caption:
Figure 1: Bottom-up annual-average methane emission inventories used as prior estimates for the inversion.

Section 3 (Results and discussion):
Figure 3 shows the ensemble mean posterior scale factors relative to the annual average prior emission estimate as described in Sect. 2.2 (left) and the corresponding averaging kernel sensitivities (right).

**Line 205: Statement "We use only high-quality retrievals …". This is misleading. If data are flagged "good" than this does not necessarily mean they are "good". Please rephrase to describe what actually has been done (We use only data with quality flag "good" (or equivalent)).**
Thank you for your comment. We rephrased to:

We use only  retrievals  with quality assessment flag equal to 1.

**Eq. (6): If I understand correctly than emission delta_x is the methane emission with unit mass per time per area whereas all other emissions in the manuscript are reported as mass per time. If this is true than please add this information or add the division by area on the right hand side of the equation explicitly.**
Thank you for your comment. You are correct that delta_x is the methane emission in units mass per area per time. This is consistent with the units used for our prior emissions as shown in Figure 1 (Mg km$^{-2}$ a$^{-1}$). All initial inverse calculations are done with these units. When we report our aggregated results, we convert these units to mass per time by multiplying by grid cell area. (We do not report gridded posterior emission rates.) We have clarified the units on delta_x and the use of area normalization during source attribution as below.

Section 2.6. (Jacobian matrix):
We assume that a perturbation of methane emissions $\Delta x_j$ in grid cell $j$ with units kg m$^{-2}$ s$^{-1}$ produces column mixing ratio enhancements $\Delta y_i$ over grid cell $i$ according to…

Section 2.8 (Source attribution):
For sectoral attribution, the rows are given by the relative, area-normalized contribution of each grid cell to a given sector in the prior emission estimate.

**Line 338 following: "We isolate anthropogenic emissions by removing contributions from wetlands and other natural sources following Sect. 2.8.". Perfect knowledge of wetland emissions compared to anthropogenic emissions is a very strong assumption given the distribution and magnitude of the wetland emissions as shown in Fig. 1. Is this considered somehow for the (surprisingly small) reported error bars (as shown in, e.g., Fig. 4)?**
Thank you for your question. We clarified that our source attribution approach does not account for the associated uncertainties in Section 2.8 (Source attribution):

The uncertainty of this method is not reflected in the reported error bounds, but the high resolution of our emission estimates decreases the influence of these assumptions relative to coarser resolution estimates.

The magnitude of the error bars results from the quality-control applied to the inversion ensemble. We clarified this in the first paragraph of Section 3 (Results and discussion):

We find 772 (421 – 1279) DOFS for the domain, where the values in parentheses here and elsewhere are the quality-controlled, eight-member inversion ensemble minimum and maximum, respectively.

Finally, our ability to separate different sources is measured by sectoral correlations as calculated from the reduced-dimension posterior error covariance matrix (as described in Section 2.8). We find low error correlations (mean error correlation coefficients less than 0.2 between all sectors), including wetlands, indicating that we can reliably separate emissions. We clarified this in Section 3.1 (CONUS sectoral emissions):

From the off-diagonal structure of $\hat{\mathbf{S}}_{K,red}$ (Eq. (8)), we find very low posterior error correlation between  all anthropogenic and biogenic sectors (mean error correlation coefficients less than 0.2), indicating that we can accurately separate sectoral emissions.

***Line 364: Please check "27.6 (22.6 – 23.9)" as mean value outside range.***
Thank you for catching this. We corrected the range of the Worden et al. emissions:

Section 3.1 (CONUS sectoral emissions):
Worden et al. (2022) found lower anthropogenic emissions of 27.6 (24.3 – 30.9) Tg a$^{-1}$ over the U.S. for 2019.

***Line 444: Please check "2.8 (2.8 – 2.9) ".***
Thanks for your comment. This is correct. We find a very narrow range of values for the Permian in our quality-controlled ensemble of inversions, reflecting the large constraint provided by TROPOMI over this region.

***Line 446: Please consider citing also Veefkind et al., 2023, studying also Permian emissions.***
Thank you for your suggestion. We added a citation to Veefkind et al., 2023, which also finds emissions consistent with our result in the Permian basin.

**Reviewer 2**

***Temporal resolution/seasonality: I couldn't find any explicit discussion of the temporal resolution of the analysis. My understanding is that a 3 hour time-step is used in the geos-chem simulations, and that the inversion is conducted for 1 full calendar year. That is to say, the analysis assumes all fluxes are static over the year and all observations, from different days with sensitivity to the same surface regions put equal (depending on winds of course) weighting on the posterior flux solution. Is that correct? It would be helpful to explicitly discuss the temporal element of the inversion. It further than would be helpful to discuss the assumptions underlying this and the implications/impacts of those assumptions. Some sources are known to have large intermittency (oil&gas) – would you expect to over/under sample those or get a reasonable average picture? It further than would be helpful to discuss the assumptions underlying this and the implications/impacts of those assumptions. Some sources are known to have large intermittency (oil&gas) – would you expect to over/under sample those or get a reasonable average picture? Do you have equal measurements across seasons? How do you treat emissions seasonal variance in the prior/posterior? Some sources, such as wetlands, have large seasonality – how do you address that? How do you consider seasonality in your landfill analysis? If you are biased toward summer observations, you might conclude high emissions as landfills have a non-insignificant temperature relationship with emissions. Is there an uncertainty term that should be added for addressing seasonal challenges?***

Thank you for your comments and questions. We broke your comments into four categories: (1) temporal resolution of the posterior emissions, (2) temporal resolution of the prior emissions, (3) temporal distribution of emissions, and (4) the effects of these three intersecting components.

(1) Temporal resolution of the posterior emissions

We optimize annual mean emissions and assume that the seasonality of the prior emissions is correct (more on this below). We clarified that we optimize annual mean emissions throughout the manuscript:

Abstract:

We quantify 2019 annual mean methane emissions in the contiguous U.S. (CONUS) at 0.25° × 0.3125° resolution by inverse analysis of atmospheric methane columns measured by the Tropospheric Monitoring Instrument (TROPOMI).

Introduction:

Here we use the reduced-rank method of Nesser et al. (2021) in an analytical inversion of 2019 TROPOMI observations to quantify annual mean emissions at 0.25° × 0.3125° resolution over North America using national emission inventories reported by the U.S., Mexico, and Canada to the UNFCCC as prior estimates.

Section 2 (Data and methods):

The $m = 2919358$ TROPOMI observations are fit to simulated GEOS-Chem concentrations to optimize annual mean methane emissions for 2019 at the 0.25° × 0.3125° GEOS-Chem resolution

Section 2.2 (Prior estimates and errors):

Figure 1 shows the annual-average prior emission estimates for different sectors.

Figure 1 caption:

Figure 1: Bottom-up annual-average methane emission inventories used as prior estimates for the inversion.

Section 3 (Results and discussion):

Figure 3 shows the ensemble mean posterior scale factors relative to the annual average prior emission estimate as described in Sect. 2.2 (left) and the corresponding averaging kernel sensitivities (right).

(2) Temporal resolution of the prior emissions

We assume that the temporal variability of the prior emissions is correct. Most emission sources have no temporal variability, except for wetlands, manure management, and rice. In addition to describing the seasonality of manure management and rice in Section 2.2 (Prior estimates and errors), we clarified the temporal resolution of the wetland prior emissions:

Prior monthly emissions for wetlands are given by the high-performance subset of the WetCHARTs ensemble version 1.3.1….

Many sources are intermittent, and this is not reflected in our prior emissions. However, errors associated with random temporal variability would be included in our estimate of the observing system errors as now described in Section 2.5 (Observing system errors):

The observing system error covariance matrix $S_O$ includes contributions from forward model, instrument, and representation errors (Brasseur and Jacob, 2017). Forward model errors include contributions from transport and from random temporal variability unresolved by the prior emissions estimate.

(3) Temporal distribution of the emissions:

There are unequal measurements across seasons in some locations but consistent observation density across the seasons over the regions of CONUS where we find a constraint from the inversion as defined by the averaging kernel sensitivities. We clarified this in the text in Section 2.4 (TROPOMI observations) and by adding a supplemental figure:

Figure 2 shows the final $m$ = 2919358 observations used for the inversion on the GEOS-Chem 0.25° × 0.3125° grid. The data are dense and seasonally consistent across high-emitting regions of CONUS (Fig. S1).

   (4)  The effects of these three intersecting components.

Temporal variability in emissions unaccounted for in the prior is assumed to average out in the inference of annual emissions. There would be an additional error term associated with this assumption, which we now acknowledge in the text in Section 2.8 (Source attribution):

> The uncertainty of this method is not reflected in the reported error bounds, but the high resolution of our emission estimates decreases the influence of these assumptions relative to coarser resolution estimates.

Seasonally distributed observations together with missing seasonality in the prior emissions could theoretically lead to biased posterior emissions. However, the analytical, reduced-rank approach will tend to optimize emissions only in areas with comparatively high annual observation density. The information content as measured by the averaging sensitivities is in part a function of observational density so that grid cells sensitive to observations only in some seasons will have comparatively small averaging kernel sensitivities. As a result, these regions will be excluded from the reduced-rank inversion, which defaults to the prior in regions with comparatively little information content. This is reflected in our inversion results, where we show almost zero information content (and, as a result, no emissions quantification) in Canada, where there are no observations in winter and some observations in summer. We clarified this in the first paragraph of Section 3 (Results):

> The high information content for CONUS reflects both the large emissions (Fig. 1) and the high density and consistency of TROPOMI observations (Fig. 2).

***Region of study:  You should be explicit that this paper analyzes emissions from onshore the 48 contiguous states – you are not constraining offshore oil/gas emission nor AK/HI.  Just a detail important for precision.  Also relevant in discussing the oil&gas sector for the entire US, as both offshore and AK are non-negligible portions of that sector.***

Thank you for your comment. Technically, offshore oil/gas emissions are included in our state vector. However, there is little constraint from TROPOMI observations for these emissions, so our reduced-rank approach defaults to the prior emissions in these areas. In addition to mentioning our focus on CONUS in the abstract and twice in the introduction, we clarified that the excluded regions are likely to represent only a small fraction of the national total:

Section 3 (Results and discussion):

> We remove emissions from Hawaii and Alaska from the GHGI total using the GHGI state estimates, which account for less than 0.5% of the national total.

Section 3.1 (CONUS sectoral emissions):

> This estimate excludes Alaska and Hawaii, which likely represent only a small (~1%) contribution to the national anthropogenic total (Miller et al., 2016; Konan and Chan, 2010).

We also addressed your concern about offshore emissions by clarifying that TROPOMI provides a constraint on most emission sources from the four largest emission sectors in Section 3.1:

> For these four sectors, we find sectoral averaging kernel sensitivities between 0.47 and 0.91, larger than the values found by Lu et al. (2022) from GOSAT and in situ data, indicating that TROPOMI constrains most of the emissions from these sources.

***Uncertainty:  I may have missed this, but how did you construct the confidence interval for the total posterior flux from the US (is that from the 8 member ensemble?)?  Please clarify. I also didn't understand how you determined uncertainty when you get to the posterior sectoral breakdown – please expand.***

Thank you for your comments. We combined your two comments on uncertainty here since the answer is the same: unless otherwise noted, confidence intervals represent the minimum and maximum of the eight-member inversion ensemble. We clarified this as follows:

Section 2.7 (Inversion ensemble):

> Unless otherwise noted,  our results give  the mean posterior emissions for the ensemble, with uncertainty ranges given by the ensemble range.

Section 3 (Results and discussion):

> We find 772 (421 - 1279) DOFS for the domain, where the values in parentheses here and elsewhere are the eight-member inversion ensemble minimum and maximum, respectively.

***Source attribution: How does the assumption on prior source distribution impact the results? If you iterate and correct %'s based on your results, would it change the sectoral? Is this type of uncertainty conveyed in the sectoral uncertainty as presented?***
Thanks for these questions! We discuss the implications of our source attribution approach in Section 2.8 (Source attribution). If I understand your point about iteration correctly, this sort of analysis would use the data multiple times to constrain emissions, violating the Bayesian assumptions implicit in our analysis. As for your question on the uncertainty estimates, we clarified that this type of uncertainty is not reflected by our ensemble-range error bounds:

> The uncertainty of this method is not reflected in the reported error bounds, but the high resolution of our emission estimates decreases the influence of these assumptions relative to coarser resolution estimates.

***Transport error: How was transport error handled/accounted for or is it not represented in uncertainty? What additional uncertainty may this add? Regional studies often find >10% uncertainty from transport error alone.***
Thanks for this question. We agree that transport error is a significant cause for concern. The residual error method (Heald et al., 2004) used here is designed to capture the total observing system errors, including contributions from transport, the instrument, and representation. We rephrased our section on observing system errors to clarify:

> The observing system error covariance matrix $\mathbf{S}_O$ includes contributions from forward model, instrument, and representation errors (Brasseur and Jacob, 2017). Forward model errors include contributions from transport and from random temporal variability unresolved by the prior emissions estimate.

***Posterior oil and gas: not necessary, but might be nice to show what your basin numbers are compared to airborne/tower studies that have been done in a bunch of the basins, even in the most recent airborne studies were ~5 years earlier. Examples include: Denver-Julesberg, Uintah, Bakken, Marcellus, among some others.***
Thank you for your suggestion! We chose not to focus on oil and gas in this paper because of the abundance of studies from our group that more directly evaluate emissions from these sources. For example, Shen et al. (2022) conduct a series of basin-by-basin inversions and show a comparison to aircraft studies in their Table S1. We point readers to this study in our Figure S3.

***Line 50 "These top-down emission estimates are most useful if they achieve high spatial resolution and maximize the information content of the observation- model system". Saying "most useful" is ill defined and misleading, as depending on question of interest high spatial resolution may not lead to most optimal result. Suggest rephrasing***
Thank you for drawing our attention to this. We clarified that high-resolution results are useful in the context of emission inventory evaluation and emission mitigation when justified with the information content of the observation-model system.

These top-down emission estimates are most useful for the evaluation of emission inventories and emission mitigation efforts if they achieve high spatial resolution  consistent with the information content of the observation-model system.

***Final sentence: you should add something like "that may be attributable to decreasing emissions between study periods" for context so the LA comparison isn't misinterpreted.***

Thank you for this suggestion. Unfortunately, we do find a low estimate compared to other studies even when we account for the study region and period (Yadav et al. (2023) quantifies emissions for 2019). We wrote:

Our urban emission estimates are in general consistent with previous top-down studies except for Los Angeles, which may be attributable in part to decreasing emissions between study periods.

***Data availability:  It would be good to include data files that have all the states and all the urban region prior/posteriors.***

Thank you for your comment and for catching this! We meant to include this data. You should now find this on the github linked under data availability under TROPOMI_inversion/inversion_data/states/ and TROPOMI_inversion/inversion_data/cities. We also added summary files for the landfill analysis. We also clarified this in the data availability section:

The prior and observational inputs for the inversion,  the posterior emissions and averaging kernel sensitivities, and summary datasets for sectors, states, cities, and landfills are available at https://github.com/hannahnesser/TROPOMI_inversion.

If we have forgotten any other important data that is sufficiently small to be stored on the github, we are happy to make additions.